# Efficient protein incorporation and release by a jigsaw-shaped self-assembling peptide hydrogel for injured brain regeneration

Atsuya Yaguchi [1,9], Mio Oshikawa[2,3,9], Go Watanabe [3,4], Hirotsugu Hiramatsu [5,6], Noriyuki Uchida[1], Chikako Hara[2,3], Naoko Kaneko[7], Kazunobu Sawamoto [7,8], Takahiro Muraoka [1,3✉] & Itsuki Ajioka [2,3✉]

During injured tissue regeneration, the extracellular matrix plays a key role in controlling and coordinating various cellular events by binding and releasing secreted proteins in addition to promoting cell adhesion. Herein, we develop a cell-adhesive fiber-forming peptide that mimics the jigsaw-shaped hydrophobic surface in the dovetail-packing motif of glycophorin A as an artificial extracellular matrix for regenerative therapy. We show that the jigsaw-shaped self-assembling peptide forms several-micrometer-long supramolecular nanofibers through a helix-to-strand transition to afford a hydrogel under physiological conditions and disperses homogeneously in the hydrogel. The molecular- and macro-scale supramolecular properties of the jigsaw-shaped self-assembling peptide hydrogel allow efficient incorporation and sustained release of vascular endothelial growth factor, and demonstrate cell transplantation-free regenerative therapeutic effects in a subacute-chronic phase mouse stroke model. This research highlights a therapeutic strategy for injured tissue regeneration using the jigsaw-shaped self-assembling peptide supramolecular hydrogel.

[1] Department of Applied Chemistry, Graduate School of Engineering, Tokyo University of Agriculture and Technology, Tokyo 184-8588, Japan. [2] Center for Brain Integration Research (CBIR), Tokyo Medical and Dental University (TMDU), Tokyo 113-8510, Japan. [3] Kanagawa Institute of Industrial Science and Technology (KISTEC), Kanagawa 243-0435, Japan. [4] Department of Physics, School of Science, Kitasato University, Kanagawa 252-0373, Japan. [5] Department of Applied Chemistry, National Yang Ming Chiao Tung University, Hsinchu 30010, Taiwan. [6] Center for Emergent Functional Matter Science, National Yang Ming Chiao Tung University, Hsinchu 30010, Taiwan. [7] Department of Developmental and Regenerative Neurobiology, Institute of Brain Science, Nagoya City University Graduate School of Medical Sciences, Aichi 467-8601, Japan. [8] Division of Neural Development and Regeneration, National Institute for Physiological Sciences, Aichi 444-8585, Japan. [9] These authors contributed equally: Atsuya Yaguchi, Mio Oshikawa. ✉email: muraoka@go.tuat.ac.jp; iajioka.cbir@tmd.ac.jp

Extracellular matrices (ECMs) are complex networks of proteins and other biomolecules, and provide a biological niche for regulating cellular responses such as proliferation, survival, and differentiation. During development, ECMs regulate the binding and release of secreted proteins, and regulate cell adhesion[1,2]. For example, secreted proteins belonging to the hedgehog family interact with and dissociate from heparan sulfate proteoglycans and generate the morphogen gradient essential for proper development[3,4]. Covalent modification of cholesterol by hedgehog family proteins is essential for the morphogen gradient[3,4] and is believed to regulate their binding to and release from heparan sulfate proteoglycans[5]. Covalently modified secreted proteins efficiently bind to and release from ECMs during injured tissue regeneration if the ECMs remain intact after injury. For example, the peptide domains of placenta growth factor-2 (PIGF-2) and laminin α subunit have high affinity to ECMs. Attachment of these peptide domains to the C-termini of growth

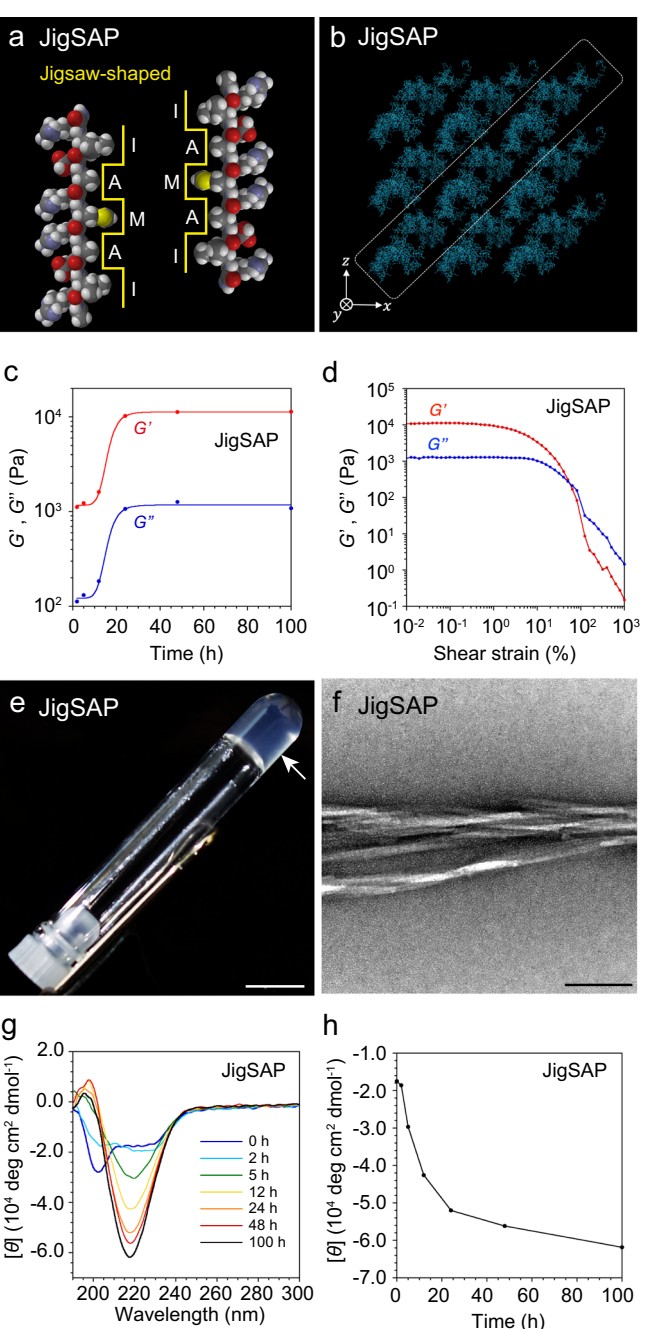

**Fig. 1 Hydrogel formation by JigSAP with micrometer-long nanofibers, robustness and high dispersibility. a** Space-filling models of Ac-RIDARMRADIR-NH$_2$ (JigSAP), showing the jigsaw-shaped hydrophobic surfaces. **b** A snapshot of supramolecular structures of JigSAP in water at 310 K obtained by all-atom MD simulation. **c** Strain-dependent storage (G', red) and loss (G'', blue) modulus profiles of JigSAP in DMEM containing 1.0 wt% HEPES at 20 °C (peptide concentration: 1.0 wt%, pH 7.4) at 100 h after incubation. **d** Time-course change in storage modulus (G', red) and loss (G'', blue) moduli of JigSAP in DMEM containing 1.0 wt% HEPES at 20 °C at 2, 5, 12, 24, 48, and 100 h after incubation (peptide concentration: 1.0 wt%, pH 7.4). G' and G'' values at the shear strain of 10$^{-1}$% were plotted. **e** Photograph of JigSAP in DMEM containing 1.0 wt% HEPES after 48 h incubation at 37 °C (peptide concentration: 1.0 wt%, pH 7.4). Arrow points to the hydrogel. Scale bar: 5 mm. **f** Transmission electron micrograph (TEM) of JigSAP. Stain: uranium acetate. Peptide concentration: 0.10 wt%, aging time: 10 s, solvent condition: 0.88 wt% aqueous solution of NaHCO$_3$, uranyl acetate concentration: 2.0 wt%. Scale bar: 100 nm. **g** Circular dichroism (CD) spectra of JigSAP in DMEM containing 1.0 wt% HEPES at 20 °C at 0, 2, 5, 12, 24, 48, and 100 h after incubation (peptide concentration: 1.0 wt%, pH 7.4). **h** Time-course change in CD signal intensity (217.6 nm) of JigSAP in DMEM containing 1.0 wt% HEPES at 20 °C at 0, 2, 5, 12, 24, 48, and 100 h after incubation (peptide concentration: 1.0 wt%, pH 7.4). Source data are provided as a Source Data file. Abbreviation; JigSAP (jigsaw-shaped self-assembling peptide).

factors resulted in the engineered growth factors enhancing signal transduction from their receptors, and enhanced tissue repair[6,7]. However, after severe tissue injury, a secreted protein must be provided along with the ECMs.

Many artificial ECMs have been created for tissue engineering and regeneration[8,9]. These artificial ECMs are divided into chemically-crosslinked polymers and supramolecularly self-assembled molecules. Various types of self-assembling peptidic materials have been reported as artificial ECMs[10–17]. Self-assembling peptide hydrogels can be widely used in clinical applications because of their cell-adhesive properties and degradability into chemically-defined molecules[18]. However, incorporating and releasing secreted proteins are generally incompatible, and development of peptidic materials capable of incorporating and releasing secreted proteins remains mostly unexplored.

Here, we show a cell-adhesive fiber-forming peptide that mimics the jigsaw-shaped hydrophobic surface in the dovetail-packing motif of glycophorin A (GYPA)[19,20], called the jigsaw-shaped self-assembling peptide (JigSAP), allowed efficient incorporation and sustained release of vascular endothelial growth factor (VEGF), and showed cell transplantation-free regenerative therapeutic effects on a subacute-chronic phase mouse stroke model.

## Results

**Material design and characterization.** AXXXA and GXXXG sequences are structural motifs found in homodimeric proteins such as glycophorin A (GYPA)[19,20]. These motifs show α-helix-to-β-strand conformational transitions, generating a jigsaw-shaped hydrophobic surface that dimerizes with dovetail packing, which in turn results in nanofiber formation through β-sheet assembly. Inspired by the dynamic self-assembling properties of these natural sequences, we designed the jigsaw-shaped self-assembling peptide JigSAP containing the AXXXA motif. JigSAP has the sequence Ac-RIDARMRADIR-NH$_2$ (Fig. 1a, ARMRA indicates the AX$^1$X$^2$X$^3$A motif; X$^1$ and X$^3$: polar amino acids, X$^2$: non-polar amino acid), with alternating hydrophilic and hydrophobic amino acid residues, generating an amphiphilic structure by exposing hydrophilic and hydrophobic surfaces in the β-sheet. The hydrophobic surface, with the sequence I–A–M–A–I, was designed based on the dovetail-packing domain

in GYPA. Molecular dynamics (MD) simulation indicated the one-dimensional self-assembly of JigSAP in an aqueous medium, suggesting nanofiber formation (Fig. 1b). JigSAP was dispersed in Dulbecco's Modified Eagle Medium (DMEM containing 1.0 wt% HEPES, pH 7.4, 20 °C). Rheological measurements of the hydrated JigSAP after 2 h incubation showed a larger storage modulus $G'$ than the corresponding loss modulus $G''$, indicating that JigSAP readily formed a hydrogel (Fig. 1c, Supplementary Fig. 1). Interestingly, the hydrogel of JigSAP showed a sharp enhancement of $G'$ over time from $1.1 \times 10^3$ Pa to $1.1 \times 10^4$ Pa between 10 and 20 h incubation, visualizing an increase in the hydrogel stiffness (Fig. 1c–e), and the JigSAP peptide hydrogel showed a thixotropic property (Supplementary Fig. 2). Transmission electron microscopic (TEM) observation of the hydrogel visualized nanofibers with 3.2-nm width on average and several micrometers long (Fig. 1f, Supplementary Fig. 3). Here, the final $G'$ ($1.1 \times 10^4$ Pa) is larger than that of conventional hydrogels consisting of peptides with alternating hydrophilic and hydrophobic amino acid residues, which typically have $G'$ values on the order of $10^2$ Pa or lower at similar concentrations[21]. Furthermore, upon increasing the shear strain, the loss modulus $G''$ of JigSAP hydrogel decreased monotonically (Fig. 1d, blue line). Typical soft glassy materials, including supramolecular gels, show weak strain overshoot, and a characteristic $G''$ profile shows a local maximum caused by temporal transitions of the fiber morphologies or alignments[22]. The monotonic decrease in $G''$ for JigSAP hydrogel results in a strain thinning profile commonly observed in covalently polymerized materials and suggests that JigSAP supramolecular nanofibers are robust. This gel-strengthening process of JigSAP was monitored by circular dichroism (CD) and infrared (IR) spectroscopies. The solution of JigSAP gave a circular dichroism (CD) spectrum with a negative Cotton effect at 202 with a shoulder between 210 and 230 nm, suggesting formation of a helix conformation (Fig. 1g). Reminiscent of proteins containing the AXXXA motif, JigSAP showed a CD spectral change, with a single minimum at 218 nm characteristic of a β-sheet structure (Fig. 1g, h). IR time-course measurement reveals a two-step structural change (Supplementary Fig. 4). After the β-sheet formation and the environmental change in the Arg side chains such as the salt-bridge formation, as indicated by the bands at 1620 and 1677 cm⁻¹, respectively, in the first step (0–5 h), the hydrogel should transform into stabler forms in the second step (12–100 h) as the spectral pattern alters without changing the secondary structure fraction. It is likely that the second step allows for the growth of nanofibers over several-μm in length. Here, the transition period in the IR spectral change corresponds to the time point showing $G'$ enhancement. Importantly, IR spectroscopic study indicates that JigSAP formed the β-sheet structure with larger than 60% fraction even at an initial stage (0 h) and the β-sheet fraction only increased slightly over the incubation time course (Supplementary Fig. 4f). This result suggests that assembly of the β-strands occurs very early and the strengthening of the hydrogel is due to consolidation of the β-sheet structures rather than a dynamic transition from the helix. Thus, it is likely that the growth of nanofibers in the late stage triggers the gelation of JigSAP. The JigSAP hydrogel after 48 h incubation involves the β-sheet structure as the amide I band at 1619 cm⁻¹ indicates (Supplementary Fig. 5, black line). The isotope labeling at Ala4 causes a new ¹³C=O band at 1590 cm⁻¹ without deforming the spectral profile of the ¹²C=O vibration at 1619 cm⁻¹ (Supplementary Fig. 5, red line). These features propose that the JigSAP hydrogel consists of the parallel β-sheet structure. Hence, the observed spectrum is consistent with a model in which JigSAP forms a parallel β-sheet structure with the JigSAP molecules packing closely without slipping[23–25]. Confocal laser scanning microscopy (CLSM) of a JigSAP hydrogel containing 1-mol%

fluorescein-labeled JigSAP showed the peptide homogeneously distributed across large fluorescent areas at the sub-millimeter scale, suggesting even dispersion of the nanofibers (Supplementary Fig. 6a). The formation of micrometer-long nanofibers by close molecular packing, structural robustness, and high dispersibility at the macro-scale, are likely responsible for the large $G'$ value for JigSAP hydrogel by increasing the crosslinking density. JigSAP showed cell adhesive properties similar to those for the well-known cell adhesive amphiphilic peptide RADA16 (Ac-RADARADARADARADA-NH₂), while both JigSAP and RADA16 did not have an actin remodeling property as fibronectin (Supplementary Figs. 7, 8). JigSAP formed hydrogel in the presence of serum, implicating the application for in vivo injection (Supplementary Fig. 9).

The geometry of the hydrophobic surface of amphiphilic peptides impacts their supramolecular properties on both the molecular and macroscopic scales. For comparison, we synthesized the flat-shaped amphiphilic peptides (FSAPs) RADA16 and 5 V (Ac-RVDVRVRVDVR-NH₂) (Supplementary Fig. 10). RADA16 is known to form a hydrogel through β-sheet formation[13,26,27], and 5 V was designed to have a flat hydrophobic surface and has a molecular weight similar to JigSAP. Here, CLSM observation of fluorescently-labeled RADA16 hydrogel showed an uneven distribution of strongly fluorescent domains and dark areas, suggesting aggregation of the peptide nanofibers (Supplementary Fig. 6b). In contrast, 5 V was predicted by MD simulations to form a non-unidirectional mesh-like structure (Supplementary Fig. 11). Indeed, 5 V remained a suspension in DMEM and did not form a hydrogel (Supplementary Fig. 12). IR spectroscopic measurement of 5 V showed amide I absorption at 1618 cm⁻¹, indicating β-sheet-type assemblies in aqueous medium (Supplementary Fig. 13). 5 V formed rod-shaped assemblies over 100 nm in length, as visualized by TEM observations (Supplementary Fig. 14).

**Protein incorporation and release.** We studied the incorporation of a full-length protein into the peptide self-assemblies by EGFP covalently tagged with JigSAP (EGFP-JigSAP) and FSAP peptides (EGFP-5V and EGFP-RADA16) at the C termini. A mammalian cell expression system was used for protein synthesis instead of a conventional *Escherichia coli* (*E. coli*) expression system because *E. coli* cannot add post-translational modifications that often regulate growth factor functions and thus are critical for biomedical applications. Peptide-tagged EGFPs were incubated with the corresponding self-assembling peptides, and the hydrogelation of the mixture was confirmed by a rheological measurement (Supplementary Figs. 15, 16). The incorporation efficiencies of the peptide-tagged EGFPs into the nanofibers were evaluated by enzyme-linked immunosorbent assay (ELISA) (Fig. 2a). Interestingly, there was a drastic difference in incorporation efficiency between JigSAP-tagged and non-tagged EGFP proteins: EGFP-JigSAP was incorporated at 93 mol%, while the incorporation of non-tagged EGFP was limited to 3 mol% (Fig. 2b). Similarly, EGFP proteins tagged with 5 V and RADA16 showed higher incorporation efficiencies into the corresponding peptide nanofibers than non-tagged EGFP (Fig. 2c, d). These results indicate the critical role of peptide tags for the incorporation of proteins. Of the three peptide-tagged EGFP proteins, EGFP-JigSAP showed a significantly higher incorporation efficiency than EGFPs with FSAP tags (EGFP-5V: 66 mol%, EGFP-RADA16: 55 mol%). It should be noted that EGFP-JigSAP incorporated in a JigSAP hydrogel showed a fluorescence spectrum characteristic of EGFP (Supplementary Fig. 17). Small-angle X-ray scattering (SAXS) profile of JigSAP essentially unchanged by incorporating EGFP-JigSAP (Supplementary Fig. 18). These results indicate retainment

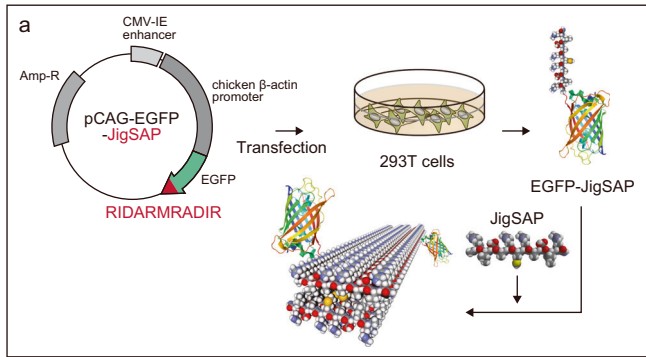

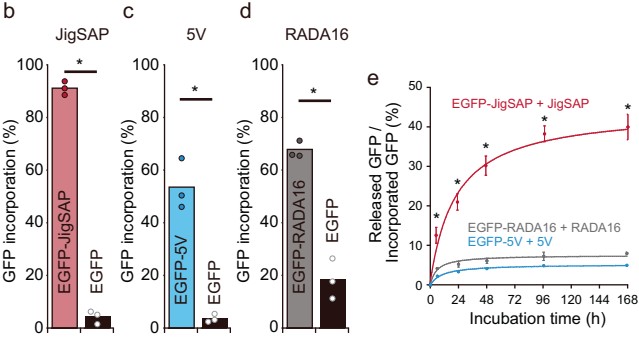

**Fig. 2 Efficient incorporation and sustained-release of JigSAP-tagged EGFP protein. a**, Experimental design: the sequence encoding the JigSAP peptide was attached to the C-terminal of *EGFP* cDNA. EGFP-JigSAP-expressing plasmids were transfected into 293 T cells. Partially purified EGFP-JigSAP proteins were incubated with JigSAP peptides. Ribbon diagram showing the EGFP structure was obtained from PDB 2Y0G. **b–d** The incorporation ratios of peptide-tagged and non-tagged EGFPs. EGFP-JigSAP (**b**, red), EGFP-5V (**c**, blue), and EGFP-RADA16 (**d**, gray) were incorporated into JigSAP, 5 V, and RADA16 peptide nanofibers, respectively. Non-tagged EGFP (**b–d**, black) was used as a control. Protein incorporation (%) = [(weight of input) – (weight of the soluble fraction)] / (weight of input) × 100 (%). $n = 3$. **e** The ratios of released GFP to incorporated GFP. EGFP-JigSAP (red), EGFP-5V (blue), and EGFP-RADA16 (gray) were released from JigSAP, 5 V, and RADA16 peptide nanofibers, respectively. Released protein / incorporated protein (%) = (total weight of the soluble fraction) / (weight of incorporated protein) × 100 (%). $n = 3$. *$P < 0.05$ (Student's *t*-test against EGFP (**b–d**) and EGFP-RADA16 + RADA16 and EGFP-5V + 5 V (**e**), two-sided). Data are mean ± SEM. Source data are provided as a Source Data file. Abbreviations; Amp-R (ampicillin resistant gene), CMV-IE (cytomegalovirus immediate early), EGFP (enhanced green fluorescent protein), JigSAP (jigsaw-shaped self-assembling peptide).

of the internal structures of the hydrogels with the native conformation of the protein bound to the supramolecular nanofibers.

Importantly, JigSAP hydrogel showed more efficient long-term release of the incorporated peptide-tagged protein than hydrogels comprising FSAPs. The release profiles for proteins from the gels were studied after the removal of non-incorporated protein by washing the hydrogels with buffer. In a time-course protein release assay, 21 mol% of EGFP-JigSAP incorporated into JigSAP nanofibers was released after 24-h incubation at 37 °C, and the percentage of released protein increased to 39 mol% after 168 h (Fig. 2e). In sharp contrast, the release of EGFP-5V and EGFP-RADA16 plateaued after 48 h incubation, with only 5 and 8 mol% release even after 168 h, respectively. As an important factor influencing the releasing rate of EGFP possessing a peptide tag from a peptide hydrogel, we compared critical aggregation concentrations (CACs) of the peptides. By monitoring fluorescence intensity changes of thioflavin T (ThT) in the presence of a

peptide at different concentrations, CACs of JigSAP and 5 V were measured to be 0.11 wt% and 0.056 wt%, respectively (Supplementary Fig. 19, CAC of RADA16: 0.023 wt%[28]). This result indicates that JigSAP is more hydrophilic than 5 V or RADA16. Thus, it is likely that the relatively high hydrophilicity of JigSAP allows for the faster protein release than RADA16 or 5 V systems. Incorporation of proteins hardly influenced the hydrogel degradation rate (Supplementary Fig. 20).

**Biomedical applications**. Given these advantageous dynamic properties of the JigSAP system, we investigated biomedical applications using vascular endothelial growth factor (VEGF) protein. VEGF enhances injured tissue regeneration by promoting angiogenesis, which is the formation of new blood vessels from existing blood vessels[29]. As anticipated, JigSAP-tagged VEGF protein (VEGF-JigSAP) was incorporated into JigSAP nanofibers more efficiently than non-tagged VEGF (VEGF-JigSAP: 63%, non-tagged VEGF: 28%; Fig. 3a). Furthermore, JigSAP tagging also influenced the release efficiency of VEGF: after 168-h incubation, 53% of the incorporated VEGF-JigSAP was released but only 5% of the non-tagged VEGF, possibly due to non-specific interaction of the protein with the self-assembled peptide (Fig. 3b). To determine whether JigSAP nanofibers incorporating VEGF-JigSAP enhance angiogenesis in vitro, we performed the human umbilical vein endothelial cell (HUVEC) tube formation assay in collagen gels. In this culture system, HUVECs exhibit a lumen-like network in the presence of VEGF[30]. The addition of 100 ng/mL (total 80 ng) of VEGF-JigSAP to the culture medium indeed resulted in HUVECs forming a lumen-like network (Fig. 3c–f). HUVECs also formed a lumen-like network when JigSAP peptides incorporating 80 ng of VEGF-JigSAP were mixed with type I collagen (Fig. 3g–j), even in the absence of VEGF-JigSAP in the culture medium. The cell number was significantly higher in the presence of 80 ng of VEGF-JigSAP in the culture medium and in the JigSAP hydrogels (Fig. 3k). It should be noted that neither JigSAP modification to VEGF nor VEGF incorporation into and releasing from JigSAP hydrogel affected VEGF bioactivity (Supplementary Fig. 21). We also examined the effect of RADA16 peptide incorporating VEGF-RADA16 (80 ng) (Supplemental Fig. 22a–c). However, VEGF-RADA16 neither promoted lumen-like network formation nor increased cell number, implicating the critical role of VEGF release from the gels. In the supernatant, enough VEGF-JigSAP to promote lumen-like network formation was detected, while VEGF-RADA16 was not (Supplemental Fig. 22d). These results suggest that the bioactivity of VEGF-JigSAP incorporated in JigSAP is due to its release from the hydrogel. Thus, self-assembling JigSAP peptides incorporating VEGF-JigSAP enhance angiogenesis in vitro.

To determine whether JigSAP-tagged proteins were released from the injected site in vivo, we injected JigSAP and RADA16 peptides incorporating EGFP-JigSAP and EGFP-RADA16, respectively into the non-injured mouse brain (Supplemental Fig. 23). At 3 h after injection, EGFP fluorescent signals were clearly detected at the injected area in both EGFP-JigSAP and EGFP-RADA16 injected brain. In contrast, at 72 h after injection, a clear EGFP signal was only detected in the EGFP-RADA16 injected brain, suggesting that EGFP-JigSAP was released from the injected site. To confirm whether JigSAP peptides does not promote striking inflammation, we immunostained for neuron marker NeuN, activated astrocyte marker GFAP, and microglia marker Iba1 at 7 days after JigSAP and RADA16 injection (Supplementary Fig. 24). Although Iba1- and GFAP-positive cells were detected in the area injecting JigSAP, the foreign body reaction level was mild and similar to the area injecting RADA16 which is already used in humans[18]. NeuN-positive cells were detected near the JigSAP-injected area similar to the RADA16-injected area (Supplementary Fig. 25).

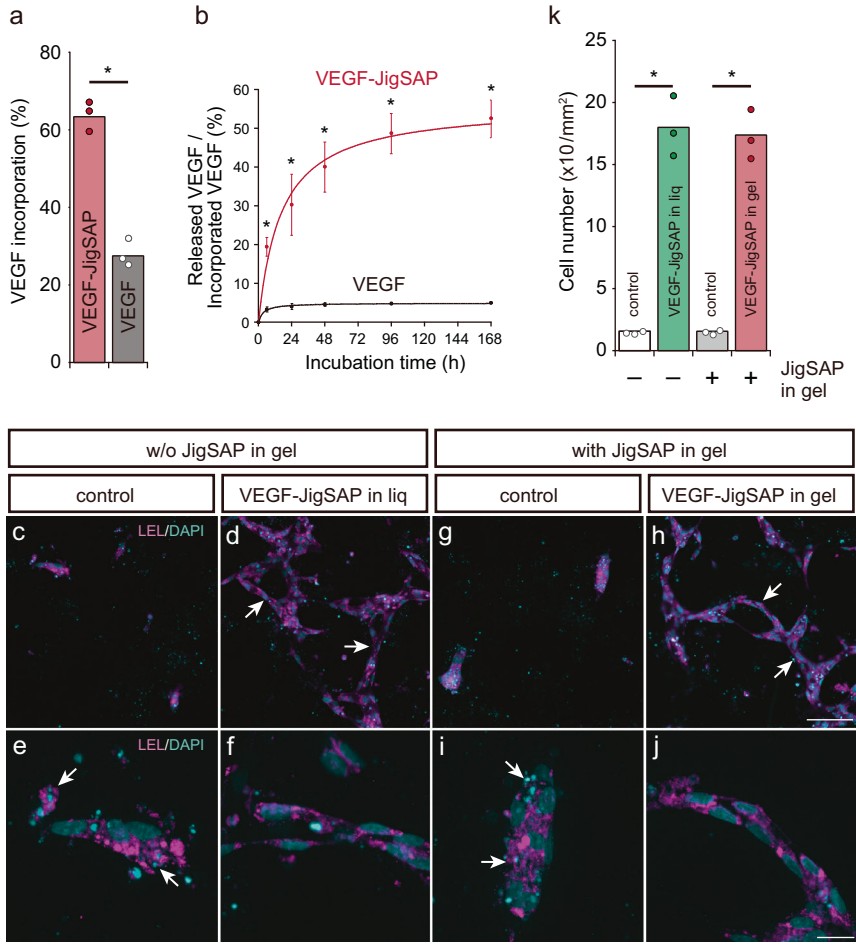

**Fig. 3 Sustained release and in vitro angiogenesis of JigSAP-tagged VEGF protein. a** The ratios of VEGF incorporation into peptide nanofibers. VEGF-JigSAP (red) and non-tagged VEGF (black) were incorporated into JigSAP peptide nanofibers. $n = 3$. **b** The ratios of released VEGF to incorporated VEGF. $n = 3$. **c–j** Endothelial cell marker *Lycopersicon esculentum* lectin (LEL) (magenta) and DAPI (cyan) images of HUVECs. Arrows in **d** and **h** show lumen-like formation by HUVECs. Arrows in **e** and **i** show condensed nuclei, a feature typical of dead cells. Scale bar: 50 μm (**c, d, g, h**) and 20 μm (**e, f, i, j**). **k** Cell scoring of HUVECs by stereology. $n = 3$. *$P < 0.05$ (Student's *t*-test against VEGF (**a, b**) and control (**k**), two-sided). Data are mean ± SEM. Source data are provided as a Source Data file. Abbreviations; VEGF (vascular endothelial growth factor), JigSAP (jigsaw-shaped self-assembling peptide), HUVEC (Human umbilical vein endothelial cell).

Next, we evaluated whether a single injection of JigSAP nanofibers incorporating VEGF-JigSAP enhances angiogenesis at the subacute-chronic phase of a mouse stroke model. We used an ischemic stroke model generated by distal middle cerebral artery occlusion (dMCAO). Seven days after dMCAO, we analyzed the behavior of the mice on an elevated wire hexagonal grid (foot-fault test), in which sensory-motor function is reflected in accurate limb placement[31] (Supplementary Mov. 1, 2). Immediately after the foot-fault test, we injected 2 μL of 1) control phosphate buffer saline (PBS), 2) 1.0% JigSAP peptide only (JigSAP only), 3) 1.6 ng of VEGF-JigSAP only (VEGF-JigSAP only), 4) 1.0% JigSAP peptide incorporating 1.6 ng of non-tagged VEGF (VEGF + JigSAP), or 5) 1.0% JigSAP peptide incorporating 1.6 ng of VEGF-JigSAP (VEGF-JigSAP + JigSAP) directly into the injured cerebral cortex as described in the Method section. Seven days after injection, we repeated the foot-fault test, followed by perfusion fixation (Fig. 4a). Injection of VEGF-JigSAP + JigSAP did not reduce the area of the injured core or of the penumbra, the area surrounding the injured core and believed important to protect from neuron death (Supplementary Fig. 26). Injection of VEGF-JigSAP + JigSAP also did not reduce the lesion size measured by the Iba1-positive area (Supplementary Fig. 27). Rather, the injection of VEGF-JigSAP + JigSAP

increased the number of endothelial cell marker laminin-positive cells at the penumbra (Fig. 4b–f, v). We examined the number of proliferated endothelial cells after injection by intraperitoneally injecting the thymidine analog EdU every 8 h for 7 days to label proliferating cells. Injection of VEGF-JigSAP + JigSAP significantly increased the number of EdU/laminin double-positive cells (Fig. 4g–u, w), suggesting enhanced angiogenesis at the penumbra. To evaluate neuroprotection, we performed Fluoro-Jade C (FJC) staining, which labels degenerating neurons[32], and NeuN staining. Injection of VEGF-JigSAP + JigSAP increased the number of NeuN-positive cells at the penumbra (Fig. 5a–j) and significantly reduced the number of FJC-positive cells (Fig. 5k–t), suggesting that it suppressed neuron death at the penumbra. Previous studies showed that VEGF enhances adult neurogenesis after brain injury[33,34]. We evaluated the effect on adult neurogenesis by examining the number of EdU/NeuN double-positive cells but observed none. Histological analysis thus suggested that a single injection of VEGF-JigSAP + JigSAP promoted angiogenesis and neuroprotection without enhancing adult neurogenesis. Finally, a foot-fault test revealed that a single injection of VEGF-JigSAP + JigSAP improved behavior whereas the other treatments had no such effect (Fig. 6a). To compare the effect of VEGF-JigSAP with VEGF-

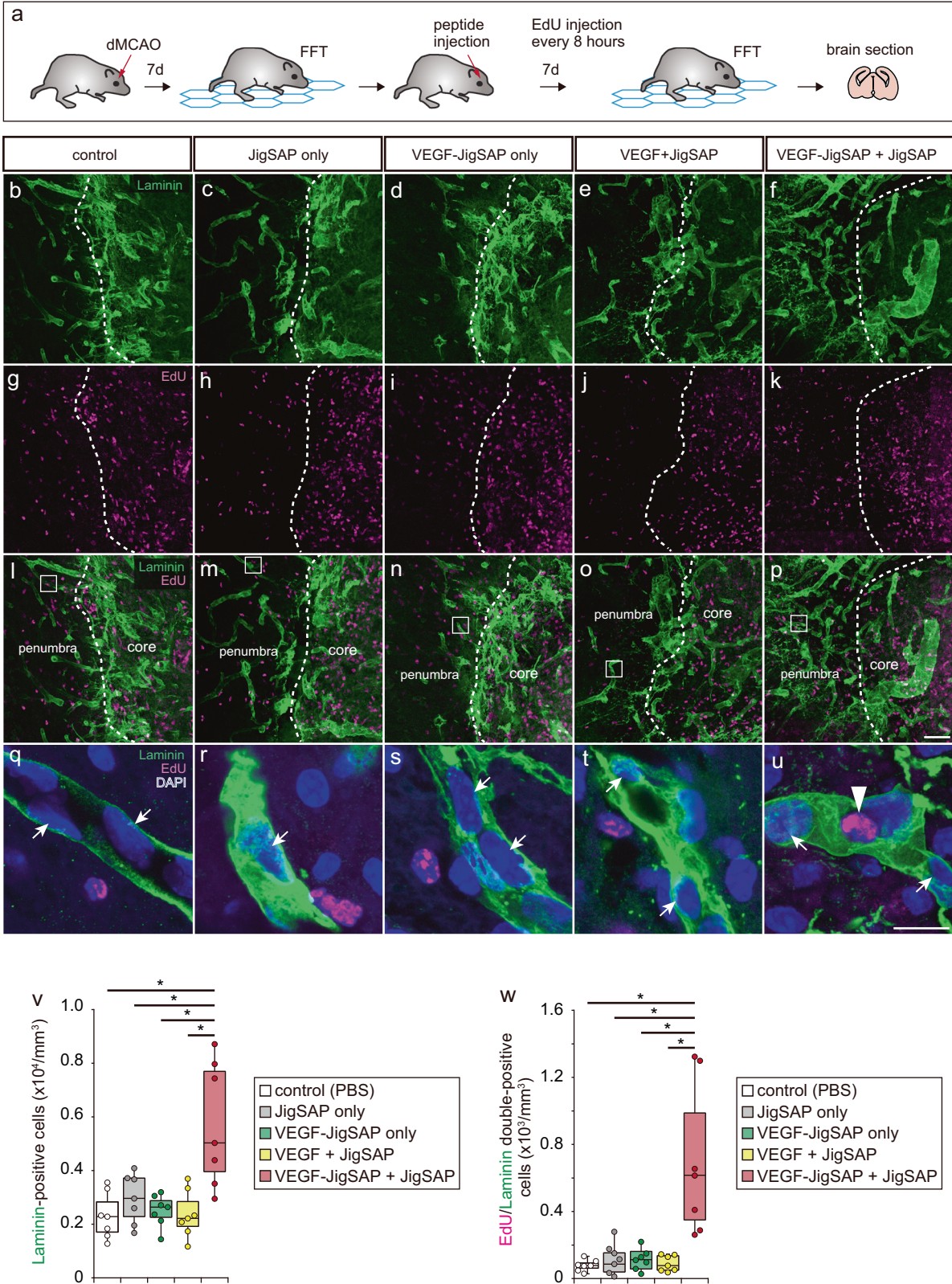

RADA16, we injected JigSAP and RADA16 peptides incorporating VEGF-JigSAP and VEGF-RADA16, respectively. In this experiment, to exclude the possibility that VEGF-JigSAP + JigSAP enhanced functional recoveries only in the dMCAO model, we used a photothrombotic model. VEGF-JigSAP + JigSAP also improved behavioral recovery in the photothrombotic model, while VEGF-RADA16 + RADA16 did not (Fig. 6b).

The behavioral recovery effect by VEGF-JigSAP + JigSAP was observed at 14 days after injection, but not at 21 days after injection because of the recovery of control mice (Supplementary Fig. 28). These results indicated that a single injection of VEGF-JigSAP + JigSAP promoted recovery of the subacute-chronic phase of this mouse ischemic stroke model at the behavioral level.

**Fig. 4 Angiogenesis of a mouse stroke model by a single injection of JigSAP nanofibers incorporating VEGF-JigSAP proteins. a** Experimental design: At 7 days after distal middle cerebral artery occlusion (dMCAO), motor coordination was assessed by a foot-fault test (FFT), then peptide mixtures were injected into the injured area. Seven days after peptide injection, the mice were retested for motor coordination and then perfusion fixed. **b–f** laminin (green), **g–k** EdU (magenta), and **l–p** laminin (green) and EdU (magenta) images at the border of the injured core and the penumbra. **q–u** High magnification images of laminin (green), EdU (magenta), and DAPI (blue) at the penumbra. **v, w** Cell scoring of laminin-positive (**v**) and EdU/laminin double-positive (**w**) cells at the penumbra. For cell counting, laminin-positive cells were counted from the non-neural lesion borders to 200 μm away. Scale bars: 50 μm (**p**) and 10 μm (**u**). *$P < 0.05$ (Student's t-test against VEGF-JigSAP + JigSAP, two-sided). $n = 7$. Box-plot elements show: center line, median; box limits, 25 and 75 percentiles; whiskers, (Q1/4–1.5IQR) and (Q3/4–1.5IQR). Source data are provided as a Source Data file. Abbreviations; dMCAO (distal middle cerebral artery occlusion), FFT (foot-fault test), EdU (5-ethynyl-2'-deoxyuridine), PBS (phosphate buffered saline), VEGF (vascular endothelial growth factor), JigSAP (jigsaw-shaped self-assembling peptide).

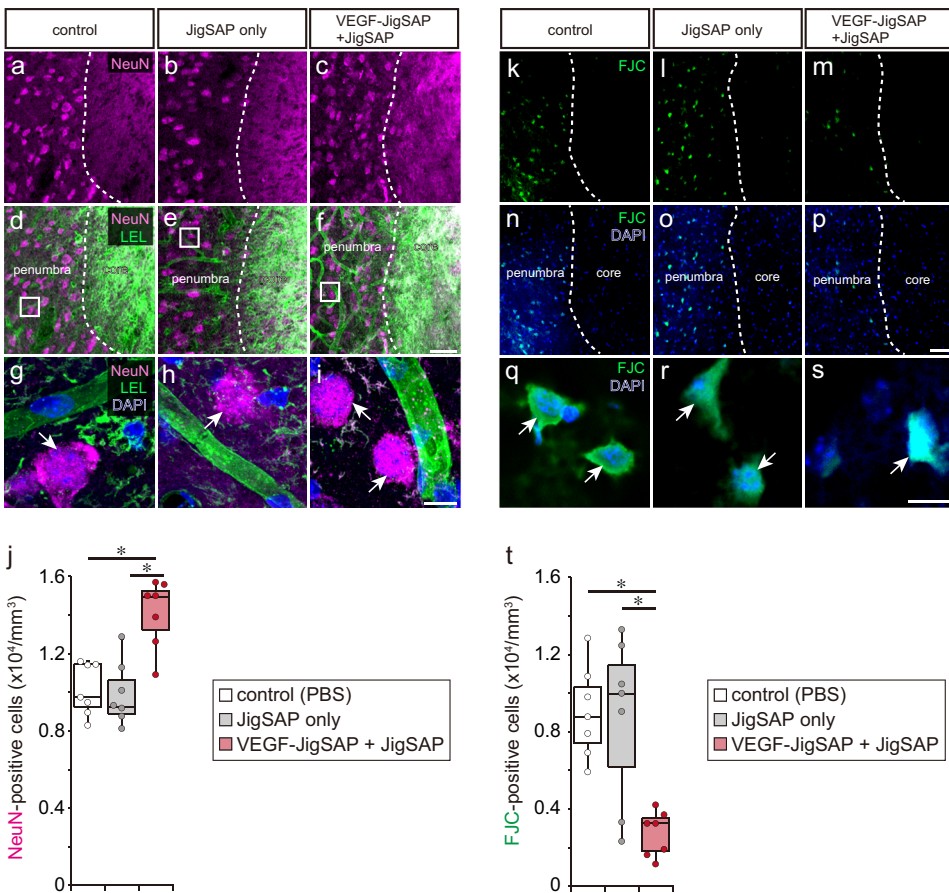

**Fig. 5 Neuroprotection of a mouse stroke model by a single injection of JigSAP nanofibers incorporating VEGF-JigSAP proteins. a–f** NeuN (magenta) (**a–c**) and NeuN (magenta)/LEL (green) (**d–f**) images at the border of the injured core and the penumbra. **g–i** High magnification images of NeuN (magenta), LEL (green), and DAPI (blue) at the penumbra. **j** Cell scoring of NeuN-positive cells at the penumbra. **k–p** FJC (green) (**k–m**) and FJC (green)/ DAPI (blue) (**n–p**) images at the border of the injured core and the penumbra. **q–s** High magnification images of FJC (green)/DAPI (blue) at the penumbra. **t** Cell scoring of NeuN-positive cells at the penumbra. For cell counting, NeuN- and FJC-positive cells were counted from the non-neural lesion borders to 200 μm away. Scale bars: 10 μm. *$P < 0.05$ (Student's t-test against VEGF-JigSAP + JigSAP, two-sided). $n = 7$. Box-plot elements show: center line, median; box limits, 25 and 75 percentiles; whiskers, (Q1/4–1.5IQR) and (Q3/4–1.5IQR). Source data are provided as a Source Data file. Abbreviations; FJC (Fluoro-Jade C), VEGF (vascular endothelial growth factor), JigSAP (jigsaw-shaped self-assembling peptide).

## Discussion

In this study, we developed JigSAP, a fiber-forming peptide bearing a jigsaw-shaped hydrophobic surface inspired by GYPA for efficient binding and release of growth factors. The association and dissociation of host and guest molecules are thermodynamic equilibrium processes, and a host–guest pair with strong affinity dissociates slowly to allow sustained release of the guest. Meanwhile, in macromolecular systems, large-scale aggregation of host–guest composites can kinetically trap the guest molecules inside the aggregates to inhibit their release. Therefore, it is integral to retain the host–guest composites with strong affinity to be dispersed for the efficient sustained release of the guest. As

seen in GYPA, JigSAP showed a helix-to-strand conformational transition, self-assembling into a parallel β-sheet structure with close molecular packing. In contrast, spectroscopic measurements of FSAPs, 5 V and RADA16 showed no conformational transitions, indicating rapid β-sheet formation. The helix-to-strand transition and close molecular packing properties of JigSAP likely allow for slow nucleation and stable growth of the self-assembly for fiber formation, which should be advantageous for efficient incorporation of tagged protein[35]. Therefore, the design of the hydrophobic surface controls the kinetics of self-assembly, in turn enhancing the efficiency of supramolecular incorporation of full-length proteins. In contrast to molecular-scale effects relating to

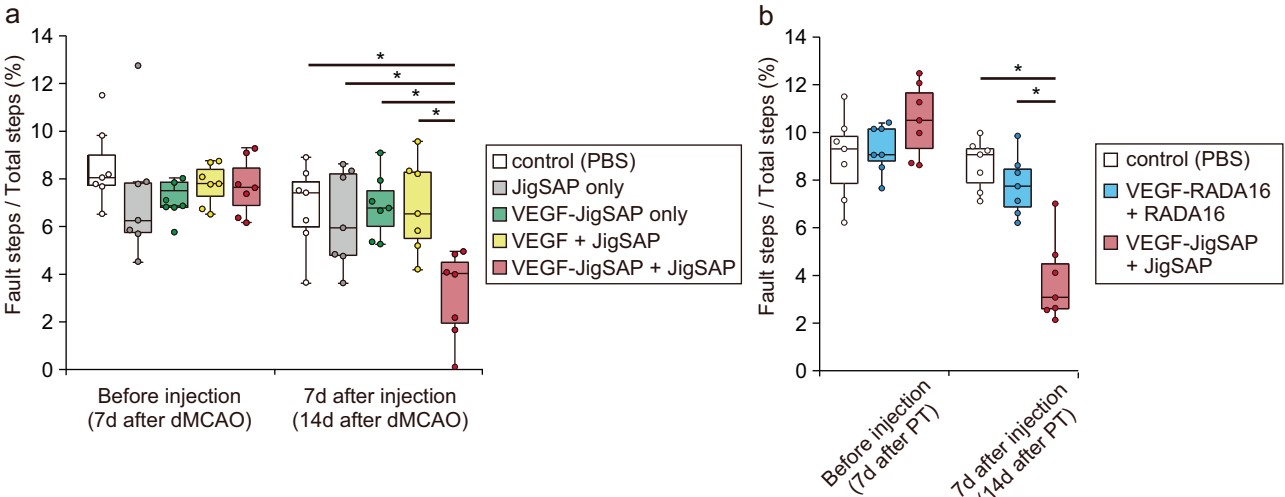

**Fig. 6 Functional recovery of a mouse stroke model by a single injection of JigSAP nanofibers incorporating VEGF-JigSAP proteins.** Ratios of fault steps to total steps during FFT. **a** dMCAO model. **b** photothrombotic model. *$P < 0.05$ (Student's *t*-test against VEGF-JigSAP + JigSAP, two-sided). $n = 7$. Box-plot elements show: center line, median; box limits, 25 and 75 percentiles; whiskers, (Q1/4–1.5IQR) and (Q3/4–1.5IQR). Source data are provided as a Source Data file. Abbreviations; dMCAO (distal middle cerebral artery occlusion), PT (photothrombotic model), PBS (phosphate buffered saline), VEGF (vascular endothelial growth factor), JigSAP (jigsaw-shaped self-assembling peptide).

the protein-incorporation process, macroscopic scale structural differences in the nanofibers likely influence the sustained protein-release properties. Specifically, sub-millimeter-scale observation under CLSM showed JigSAP nanofibers distributed widely and homogeneously, whereas RADA16, a fiber-forming FSAP, showed an uneven distribution of the nanofibers and aggregate formation. Since incorporated proteins can be trapped inside the aggregates, the wide and homogenous distribution of nanofibers as well as relatively high hydrophilicity seen with JigSAP should aid the efficient release of incorporated proteins. It should be noted that, even though the proteins tagged with Jig-SAP showed more efficient releasing from JigSAP hydrogels than unmodified proteins, their releases plateaued halfway. The incomplete releases suggest that complexation between the tagged-proteins and the peptide assemblies partially contains non-specific adsorption, although the contribution should be smaller than that formed between the non-tagged proteins and the peptides. Here, the self-assembled JigSAP showed Zeta potential of +23.3 mV (Supplementary Fig. 29). Since VEGF is also cationic (pI = 8.5)[36], it is likely that the non-specific adsorption is mainly due to hydrophobic interaction rather than electrostatic complexation. The tagging with JigSAP peptides should mitigate non-specific adsorption of VEGF onto the peptide assemblies, which serves as an advantage of this hydrogel system for preserving protein structure and controlling the release. It should be also mentioned that the hydrogel degradation rate was hardly influenced by protein incorporation.

Continuous VEGF administration using an infusion pump after brain stroke enhances angiogenesis, neuroprotection, and neurological recovery[33,37]. In this study, we found that a single injection of JigSAP peptide incorporating VEGF-JigSAP enhanced angiogenesis and neuroprotection, as expected from previous studies[33,37]. The mechanism by which VEGF suppresses neuron death after ischemic brain stroke remains unknown. In in vitro ischemic studies, VEGF directly suppressed the death of neurons expressing VEGFR2 in the cerebral cortex, the cerebellum, and the retina[38–40] whereas in vivo, newly generated blood vessels stimulated by VEGF may produce neuron survival factors. Future studies using neuron- and endothelial cell-specific conditional VEGFR2 knockout mice are required to elucidate the

mechanism of the neuroprotective role of VEGF. VEGF also enhances neurogenesis after ischemic brain stroke[33,34]. However, we did not observe EdU/NeuN double-positive cells after the injection of VEGF-JigSAP with JigSAP peptides, suggesting that neurogenesis is dispensable for functional recovery, at least in this mouse dMCAO model. Therefore, the neuroprotective role of VEGF may be crucial for functional recovery. A single injection of chemically-crosslinked heparin nanoparticle gels loaded with VEGF also promoted regeneration after ischemic brain stroke at the behavioral level[41]. Unlike chemical crosslinking materials, which degrade into chemically-undefined molecules in vivo, self-assembling peptide hydrogels degrade into chemically-defined peptides. Thus, jigsaw-shaped self-assembling peptides could be superior to chemically-crosslinked polymers for clinical applications of sustained protein release.

The biotin sandwich method was used in a seminal study on sustained protein release from self-assembling peptides, resulting in functional recovery from myocardial infarction[42]. Biotinylated insulin-like growth factor 1 (IGF-1) and biotinylated RAD16-II peptide (RARADADARARADADA) were non-covalently coupled via streptavidin and achieved sustained release of IGF-1. However, single injection of biotin-sandwich self-assembling peptide did not enhance functional recovery but rather required cell transplantation of neonatal cardiac myocytes with biotin-sandwich self-assembling peptide. Another approach to sustained release is self-assembling peptides with attached peptide tags exhibiting high affinity to growth factors[43,44]. Compared with these technologies, the method reported here is versatile as it involves simply mixing synthesized JigSAP peptides and genetically engineered JigSAP-attaching proteins at neutral pH. An alternative approach involves using bioactive minimal amino acid sequences attached to self-assembling peptides. For example, KLTWQELYQLKYKGI (QK) peptide was developed to promote endothelial cell proliferation by binding to VEGFR1/2[45]. However, QK peptides attached to JigSAP did not promote angiogenesis either in vitro (Supplementary Fig. 30) or in vivo (Supplementary Fig. 31), nor functional recovery (Supplementary Fig. 32) in dMCAO mice, at least in our biological assay, implying that attachment of a bioactive peptide to self-assembling peptides requires more complicated molecular design to exhibit

bioactivity. In contrast, the technology developed here requires the simple design of various types of full-length proteins incorporating into and releasing from self-assembling JigSAP peptides. This method can be broadly applied to the sustained release of various full-length growth factors, as well as to antibody drugs and cell transplantation scaffolds.

## Methods

**Ethical regulation**. All animal experimental procedures were approved by the Animal Experiment Committee of Tokyo Medical and Dental University (A2021-172C2).

**Reagents, cells, and mice**. Acetic anhydride was purchased from Kanto Chemicals. Acetonitrile, *N,N*′-dimethylformamide (DMF), Et₂O, *N*-methyl-2-pyrrolidone (NMP), piperidine, and trifluoroacetic acid (TFA) were purchased from Kishida Chemical (Tokyo, Japan). *N,N*-Diisopropylethylamine (DIEA) was purchased from Nacalai Tesque (Kyoto, Japan). TNBS Test Kit, triisopropylsilane (TIS), 5-ethynyl-2′-deoxyuridine (EdU), and 4-pentynoic acid were purchased from Tokyo Chemical Industry (Tokyo, Japan). Fmoc-Ala-OH•H₂O, Fmoc-Arg(Pbf)-OH, Fmoc-Asp(OtBu)-OH, Fmoc-Gln(Trt)-OH, Fmoc-Glu(OtBu)-OH H₂O, Fmoc-Ile-OH, Fmoc-Lys(Boc)-OH, Fmoc-Met-OH, Fmoc-Thr(tBu)-OH, Fmoc-Trp(Boc)-OH, Fmoc-Tyr(tBu)-OH, Fmoc-Val-OH, Fmoc-NH-SAL Resin, 1-[bis(dimethylamino)methylene]-1*H*-benzotriazolium 3-oxide hexafluorophosphate (HBTU), and 1,2,3-benzotriazol-1-ol monohydrate (HOBt•H₂O) were purchased from Watanabe Chemical Industries (Hiroshima, Japan). Ultrapure water for HPLC (filtered through a 0.22 μm membrane filter, >18.2 MΩ cm) was purified in Purelab DV35 of ELGA (Buckinghamshire, UK). Nhe I, Not I, and Xho I were purchased from New England Biolabs (MA, U.S.A.). 4′,6-Diamidino-2-phenylindole (DAPI) and bovine serum albumin (BSA) were purchased from Sigma (MO, U.S.A.). cOmplete EDTA-free was purchased from Roche (Upper Bavaria, Germany). Dulbecco's Modified Eagle Medium (DMEM, high glucose), Medium 199, fetal bovine serum (FBS), penicillin-streptomycin, GlutaMAX, 1-Step Ultra TMB-ELISA Substrate Solution, Alexa Fluor 647 Azide Triethylammonium Salt, and Alexa Fluor-conjugated secondary antibodies were purchased from Thermo Fisher Scientific (MA, U.S.A). Type I collagen and a mouse anti-NeuN antibody (MAB377) were purchased from Merck (Darmstadt, Germany). TransIT-LT1 Transfection Reagent, Endothelial Cell Growth Medium 2 (ECGM2), HUVECs were purchased from TaKaRa (Shiga, Japan). A chicken anti-GFP antibody (ab19370), a rabbit anti-GFP antibody (ab290), and a rabbit anti-laminin (ab11575) were purchased from Abcam (Cambridge, U.K.). An anti-mouse VEGF antibody (AF-493-NA), biotinylated anti-mouse VEGF (BAF493), and mouse recombinant VEGF (493-MV) were purchased from R&D Systems (MN, U.S.A.). VECTASTAIN Elite ABC Standard Kit, recombinant GFP (MB-0752), and DyLight 488 Conjugated *Lycopersicon esculentum* Lectin (488-LEL) were purchased from Vector Laboratories (CA, U.S.A.). Biotinylated goat anti-rabbit IgG (111-065-003) was purchased from Jackson ImmunoResearch (PA, U.S.A.). Fluoro-Jada C Ready-to-Dilute Staining Kit was purchased from Biosensis (Thebarton, Australia). 293 T cells (RCB2202) were provided by the RIKEN BRC through the National Bio Resource Project of the MEXT/AMED Japan. C57BL/6 J mice were purchased from Charles River Laboratory Japan (Kanagawa, Japan).

**Peptide synthesis**. Peptides were synthesized by Fmoc solid-phase peptide synthesis. A condensation-reagents cocktail of HBTU (3.05 g, 8.04 mmol) and HOBt•H₂O (1.25 g, 8.16 mmol) in DMF (16 mL), a mixture of DIEA (2.75 mL) and NMP (14.25 mL), and a cleavage cocktail of TIS (62.5 μL), TFA (2.375 mL), and water (62.5 μL) were prepared just prior to the synthesis. Fmoc-NH-SAL Resin (0.10 mmol) in a polypropylene tube, LibraTube of HiPep Laboratories (Kyoto, Japan) was soaked in DMF (2 mL) over 3 h at 25 °C. After removal of DMF, piperidine in DMF (20%, 2 mL) was added and mixed with a vortex device for 1 min. After the reaction solution was removed, piperidine in DMF (20%, 2 mL) was added and the reaction tube was shaken for 10 min at 25 °C on a PetiSyzer of Hipep Laboratories. After removal of the reaction solution, the resin was washed with DMF (2 mL, 5 times), CH₂Cl₂ (2 mL, 3 times) and DMF (2 mL, 3 times). To the resin was added Fmoc-protected amino acid (0.30 mmol) dissolved in the condensation-reagents cocktail (700 μL) and the mixture of DIEA and NMP (700 μL). After shaking for 20 min at 25 °C, the reaction solution was removed and the resin was washed with DMF (2 mL, 5 times), CH₂Cl₂ (2 mL, 3 times) and DMF (2 mL, 3 times). The Fmoc deprotection reactions with piperidine and coupling reactions of Fmoc-protected amino acid were repeated following the designed sequence. After the final Fmoc deprotection reaction and washing, to the resin was added acetic anhydride in CH₂Cl₂ (25%, 2 mL) and the reaction tube was shaken for 10 min at 25 °C. After removal of the reaction solution, the resin was washed with CH₂Cl₂ (2 mL, 3 times), DMF (2 mL, 5 times), CH₂Cl₂ (2 mL, 5 times) and CH₃OH (2 mL, 5 times). To the resin was added the cleavage cocktail (2.5 mL) and the reaction tube was left to stand for 90 min at 25 °C with gentle shaking every 30 min. The solution was collected into a polypropylene centrifuge tube by filtration. The reaction tube was rinsed with TFA (500 μL, 3 times), which is also collected by filtration. To the centrifuge tube was added Et₂O (40 mL) and the tube was mixed on a vortex device for 1 min and centrifuged at 4 °C (3500 × g, 5 min)

on a micro refrigerated centrifuge Model 3700 of Kubota (Tokyo, Japan), followed by removal of the supernatant liquid. After repeating this process for 3 times, the peptide was dried under vacuum over 2 h at 25 °C, dispersed in water and lyophilized by FDU-1200 of EYELA (Tokyo, Japan) attached with GLD-051 oil rotary vacuum pump of ULVAC (Tokyo, Japan). The synthesized peptides were purified by semi-preparative high performance liquid chromatography (HPLC) performed on PU-4086-Binary pump, UV-4075 detector and CHF122SC fraction collector of JASCO (Tokyo, Japan) attached with TA12S05-2520WX Actus Triart column (20 mmφ × 250 mm) of YMC (Tokyo, Japan) with a flow rate of 18.9 mL min⁻¹. Gradient profile of the semi-preparative HPLC: water/acetonitrile = 90/10 (0 to 10 min), 40/60 (30 min), linear gradient between 10 and 30 min. Successful isolation of the peptides was checked by analytical HPLC performed on PU-4180-pump and UV-4075 detector JASCO (Tokyo, Japan) attached with TA12S05-2546WT Triart C18 column (4.6 mmφ × 250 mm) of YMC (Tokyo, Japan) with a flow rate of 1.0 mL min⁻¹. Gradient profile of the semi-preparative HPLC: water/acetonitrile = 90/10 (0 to 5 min), 40/60 (30 min), linear gradient between 10 and 30 min (Supplementary Fig. 33). The synthesized peptides were characterized with matrix-assisted laser desorption/ionization time-of-flight mass spectrometry (MALDI-TOF MS) performed on autoflex speed spectrometer of Bruker (Bremen, Germany) in a reflector positive mode with 2,5-dihydroxybenzoic acid as a matrix (Supplementary Fig. 34). The collected samples were lyophilized and stored at –20 °C or –80 °C.

MALDI-TOF MS (2,5-dihydroxybenzoic acid, reflector positive,): *m/z* calculated for Ac-RIDARMRADIR-NH₂ + H⁺ ([JigSAP + H]⁺, C₅₇H₁₀₅N₂₄O₁₆S⁺): 1413.786, found: 1413.782; *m/z* calculated for Ac-RVDVRVRVDVR-NH₂ + H⁺ ([5 V + H]⁺, C₅₇H₁₀₅N₂₄O₁₆S⁺): 1409.845, found: 1409.866; HC≡CCH₂CH₂C(=O)-RIDARMRADIR-NH₂ + H⁺ ([Alkyne-JigSAP + H]⁺, C₆₀H₁₀₇N₂₄O₁₆S⁺): 1451.802, found: 1451.838; Ac-KLTWQELYQLKYKGIRIDARMRADIR-NH₂ + H⁺ ([QK-JigSAP + H]⁺, C₁₄₉H₂₄₆N₄₅O₃₈S⁺): calculated 3305.842, found: 3305.670.

**Gelation procedure**. A lyophilized sample after HPLC purification was dissolved in water. Addition of 100 mM NaOH aq. to the aqueous solution (pH ~ 11) afforded precipitates. The precipitates were collected by centrifugation at 4 °C (3500 × g, 5 min) on a micro refrigerated centrifuge Model 3700 of Kubota, dissolved in water again, and lyophilized by FDU-1200 of EYELA attached with GLD-051 oil rotary vacuum pump of ULVAC. The obtained lyophilized peptide powder (2.5 mg) was dissolved in 247.5 μL DMEM buffer containing 4.0 mM HEPES and 1× penicillin (pH 7.4) and incubated at 37 °C under 5% CO₂ for 24 h. For incorporation and sustained release, and HUVEC tube assay, the lyophilized peptide powder (1.0 mg) was dissolved in 50 μL DMEM buffer and mixed with 50 μL of peptide-tagged protein (45.5 ng for EGFP and 40.0 ng for VEGF) in DMEM buffer. After 24 h, these assays were performed. For in vivo injection, these conjugates were immediately injected into the brain after the mixture.

**Computational details**. All-atom molecular dynamics (MD) simulations were carried out by using GROMACS 2016.6. For each system, 90 peptide and about 38,000 water molecules and 180 chloride ions were randomly inserted in the initial cubic simulation box with sides of length 11 nm by referring to the previous study[27]. The relaxation runs at 250 K and 310 K for 2 ns under the periodic boundary conditions were successively carried out after the steepest descent energy minimization. After the relaxation runs, the equilibration MD runs were performed at 1 bar and 310 K for 300 ns under the periodic boundary conditions. The Amber ff99SB-ILDN force field[46] was used for the force field parameters of the peptides and ions and the TIP4P-EW model was used for the water molecules. During the relaxation runs, carbon, nitrogen, oxygen, and sulfur atoms of the peptides were restrained to their initial position with a force constant of 1,000 kJ·mol⁻¹·nm⁻². The velocity-rescaling[47] and Berendsen barostat[48] were used to keep the temperature and pressure of the system with relaxation times of 0.2 and 2.0 ps, respectively. The equilibration runs were performed using the Nosé-Hoover thermostat[49–51] and Parrinello-Rahman barostat[52] with relaxation times of 1.0 and 5.0 ps. The all-bonds connected to hydrogen atoms were constrained with the LINCS algorithm[53] and the time step was set to 2 fs. The smooth particle-mesh Ewald method[54] was used to calculate the long-range Coulomb interactions. The real space cutoff and the grid spacing are 1.4 and 0.30 nm, respectively.

**Rheology measurements**. Rheology measurements were conducted with rotational rheometer Kinexus lab+ of Malvern Panalytical (Malvern, UK) attached with Peltier plate cartridge and 20 mmφ convex plate and 20 mmφ parallel plate geometries (PU20 and PLS20). 250 μL of a sample was loaded and sandwiched by the plates with a gap of 0.2 mm. Storage and loss moduli (G′ and G″) were obtained by amplitude sweep measurements at 1.0 Hz at strains from 0.1 to 1000% at 20 °C. G′ and G″ were taken at plateau values in the linear viscoelastic region. Replication of the rheological data was confirmed by measuring independently-prepared three samples, which showed essentially identical profiles (Supplementary Fig. 35). The data generated in this study are provided in the Source Data.

**Circular dichroism (CD) spectrometric measurements**. CD spectra were recorded on J-1100 CD spectrometer of JASCO (Tokyo, Japan) with PTC-514 peltier temperature controller. 150 μL of a sample was loaded into a quartz

assembly cell AB20-UV-0.1 of GL Sciences (Tokyo, Japan) with 0.10-mm optical path length. The data generated in this study are provided in the Source Data.

**Infrared (IR) spectroscopic measurements**. IR absorption spectra were recorded using an FT/IR–6100 Fourier transform infrared spectrometer of JASCO (Tokyo, Japan). 15 µL of a hydrated sample was loaded into a CaF$_2$ cell with 10-µm optical path length (Biocell, Biotools, FL, USA). Each spectrum was measured with 512-times accumulation at a spectral resolution of 4 cm$^{-1}$. The data generated in this study are provided in the Source Data.

**Fluorescence spectroscopic measurements**. Fluorescence spectra were recorded using RF-6000 spectrometer of Shimadzu (Kyoto, Japan). 200 µL of a solution or gel sample was loaded into a quartz cell with 10-mm optical path length (18-F/Q/10, Starna Scientific, Hainault Essex, UK). Fluorescence spectra of samples containing EGFP were measured upon excitation at 488 nm. Fluorescence spectral measurements for evaluation of the critical aggregation concentrations were performed with samples containing 25 µM thioflavin T (ThT) upon excitation at 440 nm, and the fluorescence intensities at 480 nm were used for the analyses. The data generated in this study are provided in the Source Data.

**Transmission electron micrographic (TEM) observations**. TEM observations were conducted with H-7600 of Hitachi (Tokyo, Japan) under 100 kV accelerating voltage. A peptide sample (1.0 wt%) dispersed in aqueous NaHCO$_3$ (8.8 wt%) solution was diluted with distilled water by 10 times prior to the observation. 5 µL of a sample was placed on a parafilm. Then, a carbon coated 400 mesh copper grid was positioned on top of the drop for 10 s and washed by a droplet of distilled water. For staining, a drop of 2 wt% uranyl acetate was placed on parafilm and the grid was positioned on top of the drop for 10 s. Excess liquid was gently removed using an absorbing paper. After air drying, the grid was submitted to TEM observation.

**Small-angle X-ray scattering (SAXS) measurements**. SAXS measurements were carried out with NANO-Viewer system (Rigaku, Tokyo, Japan) equipped with Dectris (Baden-Daettwil, Switzerland) PILATUS 100k detector (Cu$_{Kα}$, λ = 1.5418 Å) and NANOPIX system (Rigaku, Tokyo, Japan) equipped with Rigaku HyPix-6000 detector (Cu$_{Kα}$, λ = 1.5418 Å) using a glass capillary with a diameter of 2.5 mm. X-ray beam diameter: 0.8 mm (NANO-Viewer) and 0.6 mm (NANOPIX), irradiation time: 15 min (NANO-Viewer) and 30 min (NANOPIX), camera length: 703.75 mm (NANO-Viewer) and 1348.75 mm (NANOPIX), measurement range (2θ): 0 to 4 degree (NANO-Viewer) and 0 to 3.5 degree (NANOPIX), standard sample: Ag Behenate.

**Confocal laser scanning microscopy (CLSM) observations**. CLSM was performed on a Leica type TCS SP8 microscope, where micrographs were recorded upon excitation at 552 nm to observe fluorescence images at 565 − 620 nm under identical settings for comparison.

**Cell adhesion assay**. The dried peptides dissolved in DMEM medium (1.0% w/v, 50 µL) were sonicated using Bioruptor (UCW-310; Cosmo Bio, Tokyo, Japan) and put on a chamber slide (Millizell EZ 8-well glass; Merck, Darmstadt, Germany). After vacuum-dried (VC-96W; TAITEC, Aichi, Japan), the peptide-coated cover-glasses were rinsed with PBS one time. NIH 3T3 fibroblasts (3.6 × 10$^5$ cells for 30 min culture and 3.6 × 10$^4$ cells for 24 h culture) were suspended in DMEM medium including 10 % FBS, plated on the peptide-coated cover-glasses, and incubated in a 5% CO$_2$ incubator. After 30 min and 24 h, cells were gently washed with PBS one time and fixed with 4% paraformaldehyde for 15 min at 25 °C. The fixed cells were washed with PBS including 0.5% Triton X-100 three times and stained with Alexa Fluor 594 Phalloidin for 30 min, and DAPI (2.0 µg/mL) for 10 min. Fluorescence micrographs were captured using a fluorescence microscope. The DAPI-positive nuclei were counted by unbiased 2D stereology (Stereo Investigator, MBF Bioscience, VT, U.S.A.). The data generated in this study are provided in the Source Data.

**Fusion protein**. To generate *pCAG-EGFP-Histag*, the *EGFP* cDNA attached with Nhe I and Not I sites was amplified from *pCAG-GFP* plasmid[55] with the following primes 5'-TTGCTAGCATGGTGAGCAAGGGCGAGGA-3' and 5'-TTGCGGCC GCCTTGTACAGCTCGTCCATG-3'. The amplicon was then inserted into the Nhe I/Not I sites of *pCAG-CST-Histag*[56]. To generate *pCAG-EGFP-Histag-RADA16*, the *6×Histag-RADA16* cDNA attached Not I and Xho I sites was generated by annealing the following oligo DNAs (5'-GGCCCATCATCATCATCA TCATCGAGCAGGAGCCCGTGCGGAGCTGTAGAGCGGACGCCAGGACA-GATGCTTAA-3' and TCGAGTTAAGCATCTGCTCTGGCGTCCGCTCTAG-CATCCGCACGGGCGTCTGCTCGATGATGATGATGATGATG-3') and then inserted into the Not I/Xho I sites of *pCAG-EGFP-Histag*. To generate *pCAG-EGFP-Histag-IAMAI* and *pCAG-EGFP-Histag-5V*, the fusion protein cDNAs attached with Nhe I and Xho I sites were amplified by PCR from *pCAG-EGFP-Histag* with the following primers 5'-TTGTCCCAAATCTGTGCGGAGCC-3' and 5'-TTGAGCTCTTACCGTATATCTGCTCGCATTCTTGCGTCGATACGATGA

TGATGATGATGATGTGCGGC-3' for IAMAI and 5'-GGAGCCGAAATCTGG-GAG-3' and 5'-TGCTCGAGTTACCGTACATCAACTCGAACTCTTACGTCCA-CACGATGATGATGATGATGATGTGCGGC for 5 V. The amplicon was then inserted into the Nhe I/Xho I site of *pCAG-EGFP-Histag*. To generate *pCAG-VEGF-Histag-IAMAI*, the fusion protein cDNAs attached with Nhe I and Xho I sites were amplified by PCR from *pCAG-VEGF-Histag*[56] with the following primers 5'-TTGTCCCAAATCTGTGCGGAGCC-3' and 5'-TTGAGCTCTTACCGTA-TATCTGCTCGCATTCTTGCGTCGATACGATGATGATGATGATGATGTGCG GC-3'. The amplicon was then inserted into the Nhe I/Xho I site of *pCAG-VEGF-Histag*. To obtain genetically engineered proteins, 293 T cells (5.2 × 10$^5$ cells) were plated in 6-well plates in 2.5 mL of DMEM with 10% FBS. One day later, the cells were transiently transfected with *pCAG-EGFP-Histag*, *pCAG-EGFP-Histag-RADA16*, *pCAG-EGFP-Histag-IAMAI*, *pCAG-VEGF-Histag*, or *pCAG-VEGF-Histag-IAMAI* using TransIT-LT1 Transfection Reagent according to the manufacturer's instructions. To obtain Histidine-tagged VEGF and VEGF-JigSAP, the conditioned medium was collected at 7 days after transfection, passed through a 0.45-µm filter, concentrated by an ultrafiltration column (Amicon Ultra 10 K; Merck, Darmstadt, Germany), displaced in the fusion protein buffer [137 mM NaCl, 2.70 mM KCl, 0.810 mM Na$_2$HPO$_4$, 0.147 mM KH$_2$PO$_4$ (pH7.4)], and kept at −80 °C before use. To obtain Histidine-tagged EGFP, EGFP-RADA16, EGFP-JigSAP, and EGFP-5V, the transfected 293 T cells were lysed in lysis buffer [20 mM Tris (pH 7.4), 150 mM NaCl, 1 mM ethylene-diamine-tetra-acetic acid (EDTA), 1% NP-40, and protease inhibitor (cOmplete EDTA-free)], and the samples were centrifuged at 15,000 ×g for 5 min. The soluble fraction was concentrated by an ultrafiltration column (Amicon Ultra 10 K, Merck), displaced in the fusion protein buffer, and kept at −80 °C before use. For rheology measurements and SAXS profiles experiments, we used the *E. coli* expression system to obtain EGFP-JigSAP and EGFP-RADA16 because a high amount of EGFP-JigSAP and EGFP-RADA16 were required. *EGFP-Histag-IAMAI* and *EGFP-Histag-RADA16* were inserted into the Xho I/Eco RI site of pRSETb vector (V35120, Thermo Fisher Scientific). Recombinant proteins were produced in *E. coli* (JM109 DE3). After the freeze-thaw cycles of the *E. coli*, the lysates were applied to NI-NTA agarose column (30210, QIAGEN). EGFP-JigSAP and EGFP-RADA16 were eluted with 200 mM imidazole, and the buffer was replaced with PBS by ultrafiltration.

**ELISA**. EGFP and VEGF quantification was done by sandwich ELISA. A chicken anti-GFP antibody (1:5000) or an anti-mouse VEGF antibody (1:1000) was incubated in a 96-well plate at 4 °C overnight. After washing the plates with TBS with 0.05% Tween 20 (TBS-Tw) and blocking the plates with 1% BSA in TBS-Tw, the proteins diluted with 1%BSA in TBS-Tw were incubated for 1 h. After washing, a rabbit anti-GFP antibody (1:5000) and biotinylated anti-mouse VEGF (1:1000) was added. The plates were incubated for 1 h, stained with VECTASTAIN Elite ABC Standard Kit, and visualized by 1-Step Ultra TMB-ELISA Substrate Solution. For EGFP quantification, biotinylated goat anti-rabbit IgG (1:500) was used after secondary antibody reaction. OD 600 was measured by ChroMate (Asahi Techneion, Tokyo, Japan). EGFP and VEGF were quantified using recombinant GFP and mouse recombinant VEGF, respectively as a standard.

**Incorporation assay**. Incorporation of the genetically engineered proteins into the self-assembling peptides was evaluated as follows. 50 µL of 2% JigSAP, 5 V, or RADA16 solution was mixed with 50 µL of Histidine-tagged EGFP-JigSAP (45.5 ng), EGFP-5V (45.5 ng), EGFP-RADA16 (45.5 ng), VEGF-JigSAP (40.0 ng), EGFP (45.5 ng), or VEGF (40.0 ng) in DMEM medium. After incubating them at 37 °C in a CO$_2$ incubator for 24 h, the hydrogels were washed with 250 µL of PBS. The concentration of EGFP and VEGF in the soluble fraction was measured by ELISA. EGFP and VEGF incorporation were calculated by the following formula. Protein incorporation (%) = [(weight of input) – (weight of the soluble fraction)] / (weight of input) × 100 (%). The data generated in this study are provided in the Source Data.

**Sustained-release assay**. JigSAP, 5 V, or RADA16 peptide hydrogels incorporated with full-length protein as described above were washes with PBS 3 times and incubated with 200 µL of 10% FBS in PBS at 37 °C and 500 rpm (CM-1000; EYELA, Tokyo, Japan). After 6, 12, 24, 72, 96, and 168 h, the soluble fractions were collected and then incubated with 200 µL of 10% FBS in PBS. EGFP or VEGF release was calculated by the following formula. Released protein / incorporated protein (%) = (total weight of the soluble fraction) / (weight of incorporated protein) × 100 (%). The data generated in this study are provided in the Source Data.

**HUVEC tube formation assay**. JigSAP and collagen gels (400 µL) were formed in 24-well plate (37 °C, 5% CO$_2$) for 1 h by mixing gelation medium [26.5 mM NaHCO$_3$, 0.2 mM GlutaMAX, 2.7 mg/mL type I collagen, and 6.2 mM NaOH in Medium-199 with or without 0.1% JigSAP and 200 ng/mL VEGF-JigSAP]. HUVECs (1.2 × 10$^7$ cells/mm$^2$) (D10008; TaKaRa) suspended in ECGM2 medium were seeded on collagen gel and incubated (37 °C, 5% CO$_2$) for 1 h. After removing the medium, JigSAP and collagen gels (400 µL) were covered on HUVECs. After 1 h (37 °C, 5% CO$_2$), ECGM2 medium (800 µL) supplemented with 10% FBS with or without 200 ng/mL soluble VEGF-JigSAP was added. After 3 days (37 °C, 5% CO$_2$), the cultured HUVECs with gels were fixed with 4% PFA. The gels were dissected

with surgical knife to expose HUVECs on the surface of the gels and then incubated with 488-LEL (1:1000) in PBS with 0.5% Triton X-100 (PBS-Tx) for 1 h to visualize endothelial cells. The nuclei were counterstained with 2 μg/mL DAPI. The images were captured using a microscope (IX73; Olympus, Tokyo, Japan) and a CCD camera (C10600-10B; Hamamatsu Photonics, Shizuoka, Japan). The data generated in this study are provided in the Source Data.

**Mouse ischemic stroke model**. Six- to Eight-week-old of C57BL/6 J female mice were used for ischemic stroke model. Mice were deeply anesthetized by spontaneous inhalation of isoflurane, and temporalis muscle was removed. dMCAO model was modified from the original method[57] and described previously[56]. Briefly, the right middle cerebral artery (MCA) was exposed by using an Ideal Micro-Drill (CellPoint Scientific; MD, U.S.A.) and the vessel was cauterized using a small vessel cauterizer (Gemini Cautery Kit, CellPoint Scientific). The hole was located at a position relative to bregma: 1.5 mm posterior and 6.5 mm lateral. The photothrombotic ischemia stroke mouse model was modified from the original method[58] and prepared by the following methods. After the skin removal, the right side of the skull was covered with the polyvinyl chloride mask with a square whole. The square hole was 3 mm transverse (relative to bregma: 0.5 mm to 3.5 mm) and 5 mm longitudinal (relative to bregma: 3.5 mm to −1.5 mm). At 5 min after intraperitoneal injection of 10 mg/mL of Rose bengal, the exposed square area was illuminated for 10 min by a halogen lamp (model L-62; Hozan, Osaka, Japan).

**JigSAP injection into the injured area**. At 7 days after dMCAO or photothrombosis, the mice were deeply anesthetized by spontaneous inhalation of isoflurane. Glass capillaries were prepared using a puller (model GD-1; Narishige, Tokyo, Japan) to obtain around 100 μm-tip. The MCA for dMCAO model and the center of the square area for the photothrombic model was exposed again by tweezers and 2 μL of PBS, 1.0% JigSAP peptides, VEGF-JigSAP (1.6 ng), 1.0% JigSAP peptides mixed with VEGF (1.6 ng), and 1.0% JigSAP peptides mixed with VEGF-JigSAP (1.6 ng) were injected into the ischemic core (1 mm deeper from the surface of the brain) for 5 min using microinjector (MO-10; Narishige). For EdU-labeling, 50 μg/g body-weight of EdU was injected intraperitoneally 21 times every 8 h for 7 days.

**Foot-fault test (FFT)**. At 1 day before FFT, mice were placed on an elevated wire hexagonal grid with 60 mm wide openings, and allowed to roam freely for 10 min as a training. At the day of FFT test, we performed the same procedure. Mouse behavior was recorded during FFT and analyzed later. A misstep was counted as a foot fault when a limb fell down into an opening in the grid. The ratio of fault step number to total step number was calculated as a percentage. The data generated in this study are provided in the Source Data.

**Immunohistostaining (IHC)**. The mouse brains were fixed by transcardiac perfusion with 4% PFA in 0.1 M phosphate buffer, and post-fixed in the same fixative overnight at 4 °C. Floating coronal sections (100-μm thick) were prepared using a vibratome sectioning system (VT1000S; Leica, Heidelberg, Germany). The resulting sections were incubated for 60 min at room temperature in blocking solution (10% normal goat serum in PBS-Tx), overnight at 4 °C with primary antibodies, and then for 2 h at room temperature with Alexa Fluor-conjugated secondary antibodies (1:500). The primary antibodies used for IHC were rabbit anti-Iba1 (1:2000), mouse anti-NeuN (1:900), and rabbit anti-laminin (1:200) antibodies. For nuclear staining, 2 μg/mL DAPI was used. EdU-incorporated cells were visualized by click reaction using Alexa Fluor 647 Azide Triethylammonium Salt after the primary antibody reaction. FJC staining was performed after secondary antibody reaction using Fluoro-Jada C Ready-to-Dilute Staining Kit.

**Cell counting using 2D and 3D stereology**. The immunopositive cells and DAPI-stained cells were counted by unbiased 2D and 3D stereology (Stereo Investigator; MBF Bioscience VT, USA) for dissociated cells and sectioned tissues, respectively. For 2D stereology, a counting frame of 100×100-μm was used in 300×300-μm matrices. For 3D stereology, a counting frame of 100×100×10-μm was used in 300×300-μm matrices in each 400-μm section. Volume measurement was performed using the program of Stereo Investigator. Fluorescence images were captured using a fluorescence microscope (IX73; Olympus) and a CCD camera (C10600-10B; Hamamatsu Photonics) and a confocal microscope (LSM900 with Airyscan 2; Carl Zeiss, Germany). Box-plot elements show: center line, median; box limits, 25 and 75 percentiles; whiskers, (Q1/4−1.5IQR) and (Q3/4−1.5IQR). The data generated in this study are provided in the Source Data.

**Statistics and reproducibility**. Rheology measurements, CD spectrometric measurements, IR spectroscopic measurements, Fluorescence spectroscopic measurements, TEM observations, SAXS measurements were performed two times independently. Cell adhesion assay and HUVEC assays were performed two times independently. Incorporation assay and sustained-release assay were performed three times independently. For animal experiments, we used 7 mice for one group. We removed the data when the injured area was not detected after dMCAO. All of the data were included for analysis and put into one graph.

**Reporting summary**. Further information on research design is available in the Nature Research Reporting Summary linked to this article.

## Data availability
The authors declare that the data that support the findings of this study are available within the paper and its Supplementary Information file. All other information is available from the corresponding authors upon reasonable request. Source data are provided with this paper.

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

## Acknowledgements

We thank Prof. Naoto Iwata (Tokyo University of Science) for fruitful discussion. We also thank Materials Analysis Division, Open Facility Center, Tokyo Institute of Technology, for their support with the structural analyses, and we are grateful to Dr. Takashi Kajitani for the PXRD measurements. This work was supported by the research fund of Kanagawa Institute of Industrial Science and Technology (to I.A.), Grant-in-Aid for Scientific Research (B) (JP17H03038 to G.W., JP19H02828 to T.M., and JP20H04528 to I.A.), Grant-in-Aid for Transformative Research Areas (B) JP21H05096 (to T.M.), Grant-in-Aid for Scientific Research on Innovative Areas (JP19H05718 to G.W.), JST CREST (JPMJCR19S4 to T.M.), the Izumi Science and Technology Foundation (to I.A.), the Asahi Glass Foundation (to T.M.), TEPCO Memorial Foundation Research Grant (Basic Research to T.M.), the Center for Emergent Functional Matter Science of National Chiao Tung University from the Featured Areas Research Center Program within the framework of the Higher Education Sprout Project by the Ministry of Education in Taiwan (to H.H.). The computations were performed using Research Center for Computational Science, Okazaki, Japan. TEM observation was conducted at Advanced Characterization Nanotechnology Platform of The University of Tokyo, supported by Nanotechnology Platform of the Ministry of Education, Culture, Sports, Science and Technology, Japan.

## Author contributions

T.M. and I.A. conceived the idea. T.M. designed the peptide molecules. A.Y., N.U. and T.M. performed chemical experiments and analysis. M.O., C.H. and I.A. performed biological experiments and analysis. H.H. performed IR measurements and analysis. G.W. performed MD simulation and analysis. N.K. and K.S. supported behavior experiments. G.W., H.H., T.M. and I.A. co-wrote manuscript.

## Competing interests

The authors declare no competing interests.

## Additional information

**Peer review information** *Nature Communications* thanks Timothy O'Shea and the other anonymous reviewer(s) for their contribution to the peer review this work. Peer reviewer reports are available.

