## [Peer Review File · Nature Communications]

Reviewers' Comments:

Reviewer #1:

Remarks to the Author:

This paper reports the jigsaw-shaped SAP exploiting α -helix-to- β -strand transition to produce a nanofibrous hydrogel when incorporated into RADA16 under physiological conditions. The Authors demonstrated controlled release of VEGF demonstrating the efficacy of the system in subacute-chronic mouse stroke models. Firstly, I would like to commend the authors on generating a fantastic body of work and presenting it at an appropriate level with an appropriate narrative for Nature Communications. This article was a pleasure to review. A key strength of the paper is the ability promoted angiogenesis and neuroprotection without enhancing adult neurogenesis

One point that the authors may wish to consider is how the protein is incorporated within the JigSAP and the RADA16 systems. I note in figure 1F that the JigSAP fibres are truncated so it is important to understand how this system interacts with the proteins compared with the nanofibrous structure of RADA16. In particular it would be good to understand how the system is forming a complex with the proteins. In other words, it would be good to elucidate the intimate association and assemblage of the components (RADA16 and JigSAP with corresponding proteins EGFP/VEGF). SEM/TEM does not provide this information, as the structures are collapsed. Other characterisation is also bulk in nature and does not show this information. I would suggest doing SAX and SANS to validate your structures and confirm how proteins are bound and what the differences are between the novel JigSAP and the well-established RADA16 system. You probably need to contrast matched one component out with D2O for example.

Overall, please make sure the tense of the article is consistent. Also, there is inconsistent use of please double check. Also some minor comments include:

1. Please add a reference (page 15, line 13) angiogenesis and neuroprotection, as expected from previous studies [adding reference here].
2. Please check the standard deviation in figure (2, d) they do seem very small. Are these actually standard error?
3. I would consider adding some more details in the methods so that the study can be reproduced. An example here would be the gender of the mice for instance.

Reviewer #3:

Remarks to the Author:

This study by Yaguchi and colleagues describes the development of a new self-assembled peptide hydrogel (called JigSAP) which they use to deliver VEGF to a model of stroke. The hydrogel presented is novel (as far as the reviewer understands) however the use of VEGF in stroke has been explored by numerous groups and has been published on extensively (some of these papers are referenced by the authors). Thus, this work is likely to be of only limited interest to readers of Nature Communications. Furthermore, after reviewing the manuscript I have several major technical issues that would need to be addressed before this manuscript would be suitable for publication. In its current state I do not think it warrants publication in Nature Communications.

Major Issues:

1. Hydrogel gelation – gelation mechanism, time and formulation considerations and their effects on the various outcome measures:
 - In their manuscript the authors describe that the mechanism of gelation of their JigSAP hydrogels is via an α -helix-to- β -strand transition of the peptide that takes around 48 hours to complete. I have concerns about how they arrived at this gelation method based on the data presented. The early time point CD spectra looks more like a random coil configuration and the later timepoint spectra is shifted too-far to the right to be a Beta-sheet configuration. The authors need to provide additional information to clarify this. Additionally, with the FT-IR spectra it is

unclear whether this was performed in the hydrated state or the dry lyophilized state? Why is there no kinetic data on the secondary structure transition by FT-IR as there is for the CD? The shift in the Amide-I peak overtime by FT-IR would provide necessary evidence of the secondary structure transition and should be provided. The authors should also perform deconvolution and second derivative analysis on these spectra to estimate the change in Beta-sheet content as gelation proceeds. This reviewer would ask that the authors provide more data and evidence for the α -helix-to- β -strand transition.

- The rheology presented for the mechanical properties of the hydrogel is presumably of the hydrogel at full gelation at 48hours (however this information is not provided). This reviewer suggests that the evolution of the mechanical properties of the sol to hydrogel transition over the 48 hour incubation period is important to included and could be readily performed by doing rheology at the same corresponding timepoints as the CD spectra data set. Furthermore, there is no data or information that describes the effect on mechanical properties of including the EGFP-JigSAP or VEGF-JigSAP into the hydrogel. This should be included.
- In the manuscript there is no mention of how long the JigSAP peptide was allowed to incubate in order to form a hydrogel before being used in the in vitro or in vivo experiments. This information should be provided. The time at which the hydrogel was used will likely have a significant effect on molecular release outcomes so it should be clarified. Furthermore, how does such a 2 day incubation at 37C prior to use effect the bioactivity of incorporated VEGF?

2. Binding, concentrations and bioactivity of VEGF in hydrogels:

- One of the major claims by the authors is binding of VEGF to their hydrogels. However, from what this reviewer understands the authors are effectively conjugating their peptide to VEGF to incorporate it into their hydrogels not binding it to a moiety on their hydrogel. The use of the term binding should be removed from the manuscript unless there is evidence that VEGF binds to their peptide which would necessitate surface plasmon resonance (SPR) experiments to demonstrate.
- The authors suggest faster release of the modified JigSAP-VEGF compared to the unmodified VEGF but they provide no evidence or explanation for why this is the case. What is causing the difference in release? Is the gel degrading or falling apart more quickly with the included JigSAP-VEGF?
- The authors should address how the modification of the VEGF with the JigSAP peptide and its incorporation and incubation into hydrogels effects bioactivity.
- The concentration of VEGF used in the in vivo experiments (0.8ng/uL in the hydrogel) is very low compared to the other studies that the authors reference that show the efficacy of VEGF in the context of stroke. In fact, it is on the order of >100X less compared to the studies they reference. Why did they choose such a low concentration of VEGF? This reviewer finds it hard to comprehend that the effects they are claiming in vivo occurred with such a low concentration of VEGF.

3. In vivo elevations

- Since the JigSAP hydrogel is a new previously untested hydrogel, the foreign body response of this material upon injection into uninjured CNS must be performed before being evaluated in the context of a disease model. The foreign body response for the JigSAP should also be benchmarked against/compared with the RADA or 5V formulations. The Foreign body response to the material in the CNS can alter molecular delivery outcomes and thus is required for this study. This analysis should include staining for astrocytes, microglia and neurons at the very least.
- The immunohistochemistry evaluations in this manuscript are completely inadequate for the conclusions made. The authors need to provide more lower and higher magnification images that show the relative differences between groups claimed in the graphs in figure 4. More histological analysis on lesion size differences as well as size and retention of hydrogels needs to be presented. Furthermore, information about where the IHC quantification is being performed must be provided. i.e. how far away from the non-neural lesion border of the lesion are the neuron counts being performed.

- To claim that the JigSAP material is a superior hydrogel carrier for this stroke treatment a group receiving a benchmarking hydrogel must be included (i.e. comparisons with the RADA hydrogel that are used as comparators in the front end of the paper should be included in the in vivo work).

Minor Issues:

- To claim the peptide is cell-adhesive the authors need to show cell morphology rather than just simply DAPI staining for their fibroblast experiments shown in Supplementary Figure 3. This reviewer would like to see a morphology stain, such as Phalloidin actin filament staining, to show the morphology of the cells to support the cell adhesion claims. This experiment needs to be longer term too, the current experiment lasted for only 30 minutes which is inadequate time to assess cell adhesion or cytotoxicity.
- For all graphs the individual data points must be overlaid on the graphs and individual n values must be included. Information on the statistical tests used must also be included. Currently, none of this is provided. These are requirements outlined by Nature communications for publication.
- The authors need to clarify how they determined the relative incorporation of tagged versus non-tagged proteins.
- More information on the in vivo experimental methods needs to be included in the methods section such as information on size of needle, rate of injections, anatomical coordinates of injection etc.
- More information is needed for the rheology testing including shear rates, strain values, and replication.
- Vendor and catalogue numbers are required for all antibodies.

Reviewer #4:

Remarks to the Author:

The authors detail the development of a jigsaw-shaped self-assembling peptide hydrogel that allows for efficient incorporation and sustained release of proteins. The peptides self-assemble into one-dimensional nanostructures in physiological conditions and subsequently form a hydrogel that allows for cell-adhesion to act as an artificial ECM for regenerative therapy. The authors illustrate the efficacy of their JigSP hydrogel in a subacute-chronic phase mouse stroke model, in which the VEGF-loaded hydrogels enhanced injured tissue regeneration. As the authors highlight, supramolecular peptides have been extensively studied for regenerative medicine; however, their system is capable of releasing secreted proteins. The significance of their system and how its design elements impart this key release feature are however not thoroughly explored. Particularly, the advantages of the JigSP design are not clearly highlighted and comparisons made to other self-assembling peptides (RADA16 and 5V) are superficial, dampening the reviewer's enthusiasm.

The following concerns also need to be addressed:

- It is unclear as to where AXXXA motif is incorporated into the JigSAP design as written in the main text; it needs to be highlighted (Results, 'Material design and characterization'). Moreover, the identity of the 'X'-labeled amino acids should be specified (polar, nonpolar, etc.) for this particular motif. The hydrophobic surface stated to be used here (IAMA I) is confusing as it suggests these amino acids are conjugated directly to each other, as opposed to there being hydrophilic amino acids in between these. This should be made more clear in the main text
- The authors need to provide purity data (such as from HPLC) and confirmation of correct molecular weight (such as MALDI-ToF or other mass spectra).
- Figure 1C caption says CD spectra represent 20C but the main text states 25C. This needs to be clarified.
- Figure 1D is not discussed in the main text after CD results, which is a more logical place to discuss the observation of peaks corresponding to beta sheet character.
- Figure 1E depicts TEM image of JigSAP nanofibers. The magnification is too low and a higher magnification image should be depicted here. Moreover, the diameter of these structures should be assessed. In addition, no information is provided regarding the sample preparation for the

nanofibers in the figure caption. What is the concentration, ageing time, solvent conditions, etc.? Supplementary Figure 1 is a more suitable magnification, but the image is blurry and needs better focusing. Moreover, it appears the filaments have some degree of twisting. The authors should comment on this observed behavior with respect to their JigSAP molecular design, which contrasts with the statement in the main text (Page 7, lines 11-12) about twisting.

- Figure 1 should also include CLSM images of 5V hydrogel or the RADA16 hydrogel image should be moved to the supplemental information alongside the other data related to the gel.

- TEM images should also be included of the RADA16 peptide assembly. Moreover, the high mag TEM of 5V (Sup. Fig. 8b) is blurry and a more focused image should be included. The sample preparation conditions for these TEM images should be included in the caption (concentration, solvent, age, etc.) and a diameter measurement should be included for the filamentous structures. The concentration of uranyl acetate for TEM imaging should also be included for the methods.

- For the incorporation of EGFP proteins into the JigSAP, 5V, and RADA16 fibers, were the two mixed with the peptides in a monomeric state or after assembly had occurred. This needs to be clarified, as assembly of the peptides into monomers may impede protein incorporation. Moreover, CD results suggest that it takes 48 hours for the formation of a hydrogel with a strong beta sheet signal at room temperature; however, for the incorporation of protein, the self-assembling peptides and proteins were mixed for 24 h at 37C. There is no experimental evidence provided that suggests that this is adequate conditions for beta sheet and hydrogel formation, which thus influences the amount of protein incorporation and its subsequent release, and thus the data collected here may not be representative of the system.

- Why is there a faster release of EGFP in the JigSP system compared to the RADA16 and 5V system? The differences likely could be related to molecular design elements, supramolecular packing, and critical micellization concentration (CMC) for the studied systems. The authors should discuss this.

- In the in vitro studies do not include comparisons of the JigSP system to the RADA16 and 5V systems for VEGF release, thus the advantages of this molecular design are not exemplified in a biomedical application context. These comparisons should be made and discussed to highlight the significance of the JigSP hydrogel system.

- The in vivo release rate of tagged proteins from the JigSP hydrogel should be included in this study. The in vitro release rates studied are likely very different than that in vivo, and the release rate will significantly impact the efficacy of the hydrogel system.

Point-by-Point Response to Reviewers

Reviewer #1

This paper reports the jigsaw-shaped SAP exploiting α -helix-to- β -strand transition to produce a nanofibrous hydrogel when incorporated into RADA16 under physiological conditions. The Authors demonstrated controlled release of VEGF demonstrating the efficacy of the system in subacute-chronic mouse stroke models. Firstly, I would like to commend the authors on generating a fantastic body of work and presenting it at an appropriate level with an appropriate narrative for Nature Communications. This article was a pleasure to review. A key strength of the paper is the ability promoted angiogenesis and neuroprotection without enhancing adult neurogenesis

We would like to thank reviewer #1 for her/his support and helpful comments and suggestions. We have made all of the changes suggested by this reviewer as listed below.

- 1) Reviewer Comment: “One point that the authors may wish to consider is how the protein is incorporated within the JigSAP and the RADA16 systems. I note in figure 1F that the JigSAP fibres are truncated so it is important to understand how this system interacts with the proteins compared with the nanofibrous structure of RADA16. In particular it would be good to understand how the system is forming a complex with the proteins. In other words, it would be good to elucidate the intimate association and assemblage of the components (RADA16 and JigSAP with corresponding proteins EGFP/VEGF). SEM/TEM does not provide this information, as the structures are collapsed. Other characterisation is also bulk in nature and does not show this information. I would suggest doing SAX and SANS to validate your structures and confirm how proteins are bound and what the differences are between the novel JigSAP and the well-established RADA16 system. You probably need to contrast matched one component out with D2O for example.”

Response: We thank the reviewer for her/his comments to improve our manuscript. Analogous to the nanofibrous structures formed with **RADA16**, **JigSAP** also forms a number of well-developed nanofibers over several- μm in length as observed by TEM (Fig. 1f in the original manuscript and Fig. 1d in the revised manuscript). It is reported that covalent attachment of a β -sheet fibrillizing peptide tag to a protein allows for its integration into self-assembling peptide nanofibers such as **RADA16** (Hudalla, T. A. *et al. Nat. Mater.* **13**, 829 (2014), cited in Ref. #37). As they also mention, the β -sheet fibrillizing peptide tag plays critical roles in the integration and influences the efficiency of incorporation. Consistent with the previous study, **EGFP-JigSAP** and **EGFP-FSAPs** showed significantly higher incorporation efficiencies into self-assembling peptide nanofibers than non-tagged **EGFP** (**FSAPs: 5V and RADA16**, Fig. 2b–2d). This result indicate that the peptide tag is the major interacting moiety with the peptide nanofibers, where it should be most likely that the peptide tag is anchored into the supramolecular nanofibers.

Figure R1-1: Small-angle X-ray scattering (SAXS) profiles of a) **JigSAP** (red line) and **JigSAP** incorporating **EGFP-JigSAP** (blue line) and b) **RADA16** (red line) and **RADA16** incorporating **EGFP-RADA16** (blue line) in DMEM containing 1.0 wt% HEPES at 20 °C (peptide concentration: 1.0 wt%, peptide tagged-EGFP concentration: 1.0×10^{-2} wt%, pH 7.4). The **JigSAP** hydrogel incorporating **EGFP-JigSAP** was incubated for 24 h before the measurement.

To investigate the internal structures of peptide hydrogels in the absence and presence of the tagged-proteins, we measured SAXS of the hydrogels. The SAXS profile of a **RADA16** hydrogel incorporating **EGFP-RADA16** was essentially identical to that of a hydrogel consisting of **RADA16** alone (**Fig. R1-1**). Importantly, hydrogels of **JigSAP** alone and a mixture of **JigSAP** and **EGFP-JigSAP** also showed analogous SAXS profiles to each other (**Fig. R1-1**). These results suggest that the internal structures of the supramolecular hydrogels are hardly influenced or degraded by incorporating the tagged-proteins.

Unfortunately, although we attempted, we could not have an opportunity to conduct SANS measurements.

It should be also important to note that **EGFP-JigSAP** incorporated in a **JigSAP** hydrogel showed a fluorescence spectrum characteristic of **EGFP**, indicating retainment of the native conformation of the protein bound to the supramolecular nanofibers (**Fig. R1-2**).

Figure R1-2: Fluorescence spectra of a) **EGFP** (black) and **EGFP-JigSAP** (blue) in DMEM containing 1.0 wt% HEPES at 20 °C (**EGFP** and **EGFP-JigSAP** concentrations: 1.0×10^{-4} wt%, pH 7.4, excitation at 480 nm). b) Fluorescence spectrum of **EGFP-JigSAP** incorporated in a **JigSAP** hydrogel in DMEM containing 1.0 wt% HEPES at 20 °C (peptide concentration: 1.0 wt%, **EGFP-JigSAP** concentration: 1.0×10^{-2} wt%, pH 7.4, excitation at 488 nm). The **JigSAP** hydrogel incorporating **EGFP-JigSAP** was incubated for 24 h before the measurement.

As a chemical property different between **JigSAP** and **RADA16** hydrogels, hydrophobicity is significant. We measured critical aggregation concentration (CAC) of **JigSAP** and compared it with the reported CAC value of **RADA16**. By monitoring fluorescence intensity changes of thioflavin T (ThT) in the presence of a peptide at different concentrations, CACs of **JigSAP** was measured to be 0.11 wt% (**Fig. R1-3**).

CAC of **RADA16** is reported as 0.023 wt% (Chen, Y. *et al. Int. J. Nanomed.* **13**, 2477–2489 (2018), cited in Ref. #30). Since an amphiphile possessing higher hydrophilicity generally shows a higher CAC value, this result indicates that **JigSAP** is more hydrophilic than **RADA16**. It is likely that the relatively high hydrophilicity of **JigSAP** allows for the faster protein release than the **RADA16** system. We added description about the SAXS, fluorescence and CAC results in the revised manuscript (pages 10 and 11) as follows and the data are shown in the Supplementary Information (Supplementary Figs. 14, 15 and 16).

Figure R1-3: Characterization of critical micelle aggregation concentration (CAC) of **JigSAP** in DMEM containing 1.0 wt% HEPES at 25 °C by monitoring ThT fluorescence intensity changes. ThT concentration: 25 μM. Excitation: 440 nm. Fluorescence: 480 nm.

“It should be noted that **EGFP-JigSAP** incorporated in a **JigSAP** hydrogel showed a fluorescence spectrum characteristic of **EGFP** (Supplementary Fig. 14). Small-angle X-ray scattering (SAXS) profile of **JigSAP** essentially unchanged by incorporating **EGFP-JigSAP** (Supplementary Fig. 15). These results indicate retainment of the internal structures of the hydrogels with the native conformation of the protein bound to the supramolecular nanofibers.”

“As an important factor influencing the releasing rate of EGFP possessing a peptide tag from a peptide hydrogel, we compared critical aggregation concentrations (CACs) of the peptides. By monitoring fluorescence intensity changes of thioflavin T (ThT) in the presence of a peptide at different concentrations, CACs of **JigSAP** and **5V** were measured to be 0.11 wt% and 0.056 wt%, respectively (Supplementary Fig. 16, CAC of **RADA16**: 0.023 wt%³⁰). This result indicates that **JigSAP** is more hydrophilic than **5V**

or **RADA16**. Thus, it is likely that the relatively high hydrophilicity of **JigSAP** allows for the faster protein release than **RADA16** or **5V** systems.”

2) **Reviewer Comment: “Please add a reference (page 15, line 13) angiogenesis and neuroprotection, as expected from previous studies [adding reference here].”**

Response: We thank the careful reading. We added the reference on **page 18** as follows.

“In this study, we found that a single injection of **VEGF-JigSAP** incorporating **JigSAP** peptide enhanced angiogenesis and neuroprotection, as expected from previous studies^{35,38}.”

3) **Reviewer Comment: “Please check the standard deviation in figure (2, d) they do seem very small. Are these actually standard error?”**

Response: We thank the careful reading. We corrected the standard errors in **Fig. 2e** and **Fig. 3b**.

4) **Reviewer Comment: “I would consider adding some more details in the methods so that the study can be reproduced. An example here would be the gender of the mice for instance.”**

Response: We thank the comments. We added more information regarding animal experiments on **page 36–38** as follows.

“Mouse ischemic stroke model

Six- to Eight-week-old of C57BL/6J **female** mice were used for ischemic stroke model. Mice were deeply anesthetized by spontaneous inhalation of isoflurane, and temporalis muscle was removed. **dMCAO model was modified from the original method⁵⁸ and described previously⁵⁷**. Briefly, the right middle cerebral artery (MCA) was exposed by using an Ideal Micro-Drill (CellPoint Scientific; MD, U.S.A.) and the vessel was cauterized using a small vessel cauterizer (Gemini Cautery Kit, CellPoint Scientific).

The hole was located at a position relative to bregma: 1.5 mm posterior and 6.5 mm lateral. The photothrombotic ischemia stroke mouse model was modified from the original method⁵⁹ and prepared by the following methods. After the skin removal, the right side of the skull was covered with the polyvinyl chloride mask with a square whole. The square hole was 3 mm transverse (relative to bregma: 0.5 mm to 3.5 mm) and 5 mm longitudinal (relative to bregma: 3.5 mm to -1.5 mm). At 5 min after intraperitoneal injection of 10 mg/mL of Rose bengal, the exposed square area was illuminated for 10 minutes by a halogen lamp (model L-62; Hozan, Osaka, Japan).

JigSAP injection into the injured area

At 7 days after dMCAO or photothrombosis, the mice were deeply anesthetized by spontaneous inhalation of isoflurane. Glass capillaries were prepared using a puller (model GD-1; Narishige, Tokyo, Japan) to obtain around 100 μ m-tip. The MCA for dMCAO model and the center of the square area for the photothrombotic model was exposed again by tweezers and 2 μ L of PBS, 1.0% **JigSAP** peptides, **VEGF-JigSAP** (1.6 ng), 1.0% **JigSAP** peptides mixed with **VEGF** (1.6 ng), and 1.0% **JigSAP** peptides mixed with **VEGF-JigSAP** (1.6 ng) were injected into the ischemic core (1 mm deeper from the surface of the brain) for 5 min using microinjector (MO-10; Narishige). For EdU-labeling, 50 μ g/g body-weight of EdU was injected intraperitoneally 21 times every 8 h for 7 days.”

Reviewer #2

This study by Yaguchi and colleagues describes the development of a new self-assembled peptide hydrogel (called JigSAP) which they use to deliver VEGF to a model of stroke. The hydrogel presented is novel (as far as the reviewer understands) however the use of VEGF in stroke has been explored by numerous groups and has been published on extensively (some of these papers are referenced by the authors). Thus, this work is likely to be of only limited interest to readers of Nature Communications. Furthermore, after reviewing the manuscript I have several major technical issues that would need to be addressed before this manuscript would be suitable for publication. In its current state I do not think it warrants publication in Nature Communications.

We would like to thank reviewer #2 for her/his support and helpful comments and suggestions. We have made all of the changes suggested by this reviewer as listed below.

1. Hydrogel gelation – gelation mechanism, time and formulation considerations and their effects on the various outcome measures:

- 1) Reviewer Comment: “In their manuscript the authors describe that the mechanism of gelation of their JigSAP hydrogels is via an α -helix-to- β -strand transition of the peptide that takes around 48 hours to complete. I have concerns about how they arrived at this gelation method based on the data presented. The early time point CD spectra looks more like a random coil configuration and the later timepoint spectra is shifted too-far to the right to be a Beta-sheet configuration. The authors need to provide additional information to clarify this. Additionally, with the FT-IR spectra it is unclear whether this was performed in the hydrated state or the dry lyophilized state? Why is there no kinetic data on the secondary structure transition by FT-IR as there is for the CD? The shift in the Amide-I peak overtime by FT-IR would provide necessary evidence of the secondary structure transition and should be provided. The authors should also perform deconvolution and**

second derivative analysis on these spectra to estimate the change in Beta-sheet content as gelation proceeds. This reviewer would ask that the authors provide more data and evidence for the α -helix-to- β -strand transition.”

Response: In response to the reviewer’s suggestions, we have performed time-course CD and FI-IR measurements of JigSAP hydrogels. Based on the updated spectroscopic data obtained by the most careful measurements using freshly prepared samples, we would like to amend the CD spectral data (Fig. 1c in the original manuscript and Fig. 1g in the revised manuscript) and some of our descriptions about the mechanism and process of the gel formation as explained below.

In the time-course CD measurement (**Fig. R2-1a**), JigSAP showed negative Cotton effect at 202 nm with a shoulder between 210 and 230 nm in the initial state. The initial CD profile resembles the CD pattern of peptides adopting a 3_{10} -helical conformation that shows negative Cotton effect in the 201–206 nm region accompanied by a pronounced negative shoulder centered at approximately 222 nm as described in a previous paper (Formaggio, F. *et al. Chem. Eur. J.* **6**, 4498–4504 (2000)). The broad shoulder may also include signals corresponding to α -helix structure. Meanwhile, the initial CD spectrum showed no negative Cotton signal at $\lambda < 200$ nm characteristic of a random coil structure. Therefore, it is more likely to mention that JigSAP adopts a helical conformation (including 3_{10} -, α -helix or both) in the initial stage. During incubation at 20 °C, the CD spectral profile of **JigSAP** changed to show a negative Cotton signal at 218 nm (**Fig. R2-1a,b**). Hence, the CD study indicates helix-to-sheet conformational transition of **JigSAP**.

Based on the spectral analysis written above, we changed “ α -helix” as the initial conformation of **JigSAP** to “helix” in the revised manuscript.

Figure R2-1: a) Circular dichroism (CD) spectra and b) time-course change in CD signal intensity (217.6 nm) of **JigSAP** in DMEM containing 1.0 wt% HEPES at 20 °C at 0, 2, 5, 12, 24, 48, and 100 h after incubation (peptide concentration: 1.0 wt%, pH 7.4).

All the IR spectra shown in this study were measured in the hydrated state using CaF₂ windows, of which the path length is fixed (Biocell; Biotools, FL). We performed time-course measurement of IR spectra of **JigSAP** as shown in **Fig. R2-2a**. Differential spectra indicated a two-step process during the gelation of **JigSAP** (**Fig. R2-2b**). In the first stage (stage 1: 0–5 h), the differential spectra showed positive peaks at 1677 and 1610 cm⁻¹ assigned to the Arg side chains and β -sheet structure, respectively. In the second stage (stage 2: 12–100 h), the differential spectra showed broad band.

Band decomposition analysis of the IR spectra of **JigSAP** was carried out by using five Gaussian bands (**Fig. R2-2c**). The observed IR spectrum at each time (black) was fitted well (red). Also, the second derivative of the fitted curve (red in **Fig. R2-2d**) reproduced the second derivative IR spectra (black in **Fig. R2-2d**) well. Hence the band decomposition was successful. The bands below and above 1640 cm⁻¹ were assigned to the β -sheet structure and the α -helix or random structures, respectively, since the amide I bands of the helical and random structures overlap above 1640 cm⁻¹.

The time-course plot of the amide I peak position showed that the gelation process of **JigSAP** involves two steps (**Fig. R2-2e**). The band area of each secondary structure was summed and plotted against the time (**Fig. R2-2f**). The fraction of β -sheet structure increased in stage 1 and reached a plateau after 12 h, while that of the other secondary structures decreased accordingly. The plot did not show remarkable changes in stage 2.

The two-step IR spectral change visualizes the nanofiber formation process of **JigSAP**. Namely, **JigSAP** forms an aggregate, possibly kinetically stable forms, in stage 1. The increased band intensity of the Arg side chains at 1677 cm⁻¹ suggests the environmental change of the side chain structure, such as the salt-bridge formation (**Fig. R2-2b**). This step accompanies the growth of the β -sheet structure because the amide I band at 1620 cm⁻¹ becomes larger. Then, the β -sheet structure is likely rearranged into thermodynamically stabler forms to allow for the growth of nanofibers over several- μ m in length in stage 2, as the spectral change covered the broad region without changing the fraction of each secondary structure.

This discussion was added to the revised manuscript (page 7) as follows and to Supplementary Information (Supplementary Fig. 3).

Figure R2-2: Results of the infrared (IR) absorption measurement of **JigSAP** in DMEM containing 1.0 wt% HEPES at 20 °C at 0, 2, 5, 12, 24, 48, and 100 h after starting incubation (peptide concentration: 1.0 wt%, pH 7.4). a) the IR absorption spectrum at each time, and b) their difference. c) Band decomposition and d) second derivative analyses of the IR spectra. Black and red lines indicate the measured IR spectrum and the fitted curve, respectively, in c) and the second derivative of the measured IR spectrum and the fitted curve, respectively, in d). In c), the blue bands are assigned to the β-sheet structures and the green bands are to the others (α-helix and random structures). e) Time-course changes of the peak position of the amide I band around 1620 cm⁻¹. f) Time-course changes of the fractions of the β-sheet structure (filled circles) and the others (open square) in the amide I band of **JigSAP** evaluated by the deconvolution analysis shown in c).

“IR time-course measurement reveals a two-step structural change (Supplementary Fig. 3). After the β-sheet formation and the environmental change in the Arg side chains such as the salt-bridge formation, as indicated by the bands at 1620 and 1677 cm⁻¹,

respectively, in the first step (0–5 h), the hydrogel should transform into stabler forms in the second step (12-100 h) as the spectral pattern alters without changing the secondary structure fraction. It is likely that the second step allows for the growth of nanofibers over several- μm in length. Here, the transition period in the IR spectral change corresponds to the time point showing G' enhancement. Thus, it is likely that the growth of nanofibers in the late stage triggers the gelation of **JigSAP**.”

2) **Reviewer Comment:** “The rheology presented for the mechanical properties of the hydrogel is presumably of the hydrogel at full gelation at 48hours (however this information is not provided). This reviewer suggests that the evolution of the mechanical properties of the sol to hydrogel transition over the 48 hour incubation period is important to included and could be readily performed by doing rheology at the same corresponding timepoints as the CD spectra data set. Furthermore, there is no data or information that describes the effect on mechanical properties of including the EGFP-JigSAP or VEGF-JigSAP into the hydrogel. This should be included.”

Response: In response to the reviewer’s suggestions, we have performed time-course rheology measurements of the **JigSAP** hydrogel at the same timepoints as the CD measurements obtained by the most careful measurements using freshly prepared samples (**Fig. R2-3**). The time-course profile of G' clearly shows sharp enhancement between **10 and 20 h** followed by plateau (G' : 1.1×10^3 Pa to 1.1×10^4 Pa). Importantly, this period showing G' enhancement corresponds to the transition point between stages 1 and 2 observed in the IR analyses. Thus, it is likely that the growth of nanofibers in the late stage triggers the gelation of **JigSAP**. We added a description and a figure about the time-course G' change to the main text (page 6, Fig. 1f).

Figure R2-3: Time-course change in storage modulus G' of **JigSAP** in DMEM containing 1.0 wt% HEPES at 20 °C at 2, 5, 12, 24, 48, and 100 h after incubation (peptide concentration: 1.0 wt%, pH 7.4). G' values at the shear strain of $10^{-1}\%$ were plotted.

“Dynamic viscoelastic measurements of hydrated **JigSAP** showed a sharp enhancement of the storage moduli G' over time from 1.1×10^3 Pa to 1.1×10^4 Pa between 10 and 20 h incubation (Fig. 1e,f).”

Rheological property of **JigSAP** hydrogel incorporating **EGFP-JigSAP** was measured (**Fig. R2-4**). **JigSAP** hydrogel incorporating **EGFP-JigSAP** showed a comparable level of G' to **JigSAP** hydrogel. We added a description and the data about the rheology to page 10 and **Supplementary Fig. 13**.

Figure R2-4: Strain-dependent storage (G' , red) and loss (G'' , blue) modulus profiles of **JigSAP** incorporating **EGFP-JigSAP** in DMEM containing 1.0 wt% HEPES at 20 °C (incubation time: 48 h, peptide concentration: 1.0 wt%, **EGFP-JigSAP** concentration: 1.0×10^{-4} wt%, pH 7.4).

“Peptide-tagged **EGFPs** were incubated with the corresponding self-assembling peptides, and the hydrogelation of the mixture was confirmed by a rheological measurement (Supplementary Fig. 13).”

- 3) Reviewer Comment: “In the manuscript there is no mention of how long the **JigSAP** peptide was allowed to incubate in order to form a hydrogel before being used in the in vitro or in vivo experiments. This information should be provided. The time at which the hydrogel was used will likely have a significant effect on molecular release outcomes so it should be clarified. Furthermore, how does such a 2 day incubation at 37C prior to use effect the bioactivity of incorporated **VEGF**?”

Response: We apologize that the important information was described in the different sections. For *in vitro* experiments, we mixed **JigSAP** and the **JigSAP**-tagged protein and incubated them for 24 hours after *G'* becomes plateau. For *in vivo* experiments, we injected immediately after mixing the peptides because gel injection using a capillary needle was extremely hard. When we injected the mixture of **JigSAP** incorporating **JigSAP-EGFP** into the non-injured brain, we could observe the cell-free area even after 72 hours (the results showing at the response to reviewer comment (8): **Fig. R2-7**), suggesting that injected materials formed gel *in vivo*. In the revised manuscript, we moved the information of gelation time for *in vitro* and *in vivo* in the “Gelation procedure” section in the methods (Page 27) as follows.

“For incorporation and sustained release, and HUVEC tube assay, the lyophilized peptide powder (1.0 mg) was dissolved in 50 μ L DMEM buffer and mixed with 50 μ L of peptide-tagged protein (45.5 ng for EGFP and 40.0 ng for VEGF) in DMEM buffer. After 24 h, these assays were performed. For *in vivo* injection, these conjugates were immediately injected into the brain after the mixture.”

As for the comment on the bioactivity of **VEGF-JigSAP** after 24 h incubation in **JigSAP**, we discussed it to the reviewer comment (6).

2. Binding, concentrations and bioactivity of VEGF in hydrogels:

- 4) **Reviewer Comment:** “One of the major claims by the authors is binding of VEGF to their hydrogels. However, from what this reviewer understands the authors are effectively conjugating their peptide to VEGF to incorporate it into their hydrogels not binding it to a moiety on their hydrogel. The use of the term binding should be removed from the manuscript unless there is evidence that VEGF binds to their peptide which would necessitate surface plasmon resonance (SPR) experiments to demonstrate.”

Response: We thank the comment. We deleted the term of “binding” in the abstract (page 2), the introduction (page 4), and the discussion (page 21).

5) **Reviewer Comment:** “The authors suggest faster release of the modified JigSAP-VEGF compared to the unmodified VEGF but they provide no evidence or explanation for why this is the case. What is causing the difference in release? Is the gel degrading or falling apart more quickly with the included JigSAP-VEGF?”

Response: In response to the reviewer’s comment, we investigated degradation of the hydrogels upon addition of the peptide-tagged proteins as a possible factor influencing the release rate of proteins. SAXS profiles of **JigSAP** hydrogels incorporating no proteins, **EGFP-JigSAP** and non-tagged **EGFP** showed essentially identical profiles, thus suggesting addition of proteins hardly causes degradation of the internal structures of the gels (**Fig. R2-5**).

As we show in Fig. 3a, incorporation efficiency of **VEGF-JigSAP** into a **JigSAP** hydrogel was significantly larger than that of non-tagged **VEGF**, suggesting an effective role of the **JigSAP** tag for the incorporation presumably as an anchoring unit. Analogously, **EGFP-JigSAP** was also incorporated into a **JigSAP** hydrogel more efficiently than non-tagged **EGFP** (**Fig. 2b**). These results suggest that the incorporation mechanism of proteins without peptide tags into the hydrogel should be different from that of the tagged-proteins, and one of the possible mechanisms of incorporation of the non-tagged proteins is non-specific adsorption. The significantly slower release of the non-tagged proteins from the hydrogel was thus likely due to the non-specific interaction of the proteins with the peptide hydrogel.

We added a description about this point to the main text (**page 10**) as follows and **Supplementary Fig. 15**.

Figure R2-5: Small-angle X-ray scattering (SAXS) profiles of a) **JigSAP** (red line) and **JigSAP** incorporating **EGFP-JigSAP** (blue line) and b) **RADA16** (red line) and **RADA16** incorporating **EGFP-RADA16** (blue line) in DMEM containing 1.0 wt% HEPES at 20 °C (peptide concentration: 1.0 wt%, peptide tagged-EGFP concentration: 1.0×10^{-2} wt%, pH 7.4). The **JigSAP** hydrogel incorporating **EGFP-JigSAP** was incubated for 24 h before the measurement.

“Small-angle X-ray scattering (SAXS) profile of **JigSAP** essentially unchanged by incorporating **EGFP-JigSAP** (Supplementary Fig. 15). These results indicate retention of the internal structures of the hydrogels with the native conformation of the protein bound to the supramolecular nanofibers.”

6) Reviewer Comment: “The authors should address how the modification of the **VEGF** with the **JiGSAP** peptide and its incorporation and incubation into hydrogels affects bioactivity”

Response: We agree, and to address this issue, we performed an ELISA assay of HUVEC supernatant (**Fig. R2-6**). These results showed that there was enough amount of **JigSAP**-tagged **VEGF** promoting HUVEC tube formation in the supernatant. Thus, the bioactivity is likely due to the released **JigSAP-VEGF** from **JigSAP** gel.

Figure R2-6: ELISA assay of the conditioned medium

As for the reviewer’s comment (3), the secreted **VEGF-JigSAP** in the conditioned medium has an enough bioactivity for HUVEC. Thus, **VEGF-JigSAP** in **JigSAP** does not lose bioactivity at least for 5 days.

We added the information to the main text (page 13) as follows and Supplementary Fig. 17d.

“In the supernatant, enough **VEGF-JigSAP** to promote lumen-like network formation was detected, while **VEGF-RADA16** was not (Supplemental Fig. 17d). These results suggest that the bioactivity of **VEGF-JigSAP** incorporated in **JigSAP** is due to its release from the hydrogel.”

7) Reviewer Comment: “The concentration of VEGF used in the in vivo experiments (0.8ng/uL in the hydrogel) is very low compared to the other studies that the authors reference that show the efficacy of VEGF in the context of stroke. In fact, it is on the order of >100X less compared to the studies they reference. Why did they choose such a low concentration of VEGF? This reviewer finds it hard to comprehend that the effects they are claiming in vivo occurred with such a low concentration of VEGF.”

Response: We appreciate the excellent comments and views. Based on our **VEGF** release assay, we predicted that 50 pg/day of **VEGF-JigSAP** is released from the

injected **JigSAP** during 5 days after injection at 0.8 ng/uL. We roughly estimate that 50 pg/uL of **VEGF-JigSAP**, which is the same concentration of HUVEC tube assay, works in the injected area. Thus, we thought 0.8 ng/uL *in vivo* is a reasonable concentration leading to the recovery of the ischemic mouse stroke model. In the revised manuscript, to describe and emphasize that the concentration in this study was lower than the previous seminal study, we added the following sentence on page 18.

“Recently, Nih et al. reported that the single injection of heparin nanoparticles binding 200 ng of VEGF promotes functional recoveries³⁹. However, in the current study, we injected **JigSAP** peptides incorporating 1.6 ng of **VEGF-JigSAP** and observed functional recoveries, suggesting that a lower dose of VEGF may be enough for functional recoveries if VEGF is slowly released from the injected site.”

3. In vivo elevations

- 8) **Reviewer Comment:** “Since the **JigSAP hydrogel** is a new previously untested hydrogel, the foreign body response of this material upon injection into uninjured CNS must be performed before being evaluated in the context of a disease model. The foreign body response for the **JigSAP** should also be benchmarked against/compared with the **RADA** or **5V** formulations. The Foreign body response to the material in the CNS can alter molecular delivery outcomes and thus is required for this study. This analysis should include staining for astrocytes, microglia and neurons at the very least.”

Response: We agree. To address this issue, we injected **JigSAP** and **RADA16** into the non-injured mouse brain and fixed it at 72 hours after injection. We performed immunostaining using the antibodies against GFAP, Iba1, and NeuN (Figs. **R2-7, R2-8**). These results suggest that **JigSAP** did not cause a remarkable inflammation response compared with the **RADA16** peptides, which FDA already approves.

Figure R2-7: Evaluation of the foreign body responses of EGFP-JigSAP and EGFP-RADA16 after brain injection.

EGFP-JigSAP and **EGFP-RADA16** was injected into the non-injured brains. At 72 h after injection, these mice were fixed. Activated microglial marker Iba1 (green) (**a, b, g, h**), activated astrocyte marker GFAP (magenta) (**c, d, i, j**), and merged DAPI (blue) images. Asterisks (**c, e, i, k**) show non-specific staining by anti-GFAP. Scale bars: 200 μm (**f**) and 20 μm (**l**).

Figure R2-8: NeuN staining after EGFP-JigSAP and EGFP-RADA16 injections.

EGFP-JigSAP and EGFP-RADA16 were injected into the non-injured brains. At 72 h after injection, these mice were fixed. Activated microglial marker Iba1 (green) (a, b), neuron marker NeuN (magenta) (c, d), and merged DAPI (blue) images. Scale bar: 20 μ m.

We added the information to the main text (page xx) as follows and **Supplementary Figs. 19 and 20**.

“To confirm whether **JigSAP** peptides did not promote striking inflammation, we immunostained for neuron marker NeuN, activated astrocyte marker GFAP, and microglia marker Iba1 at 72 h after **EGFP-JigSAP** and **EGFP-RADA16** injection (Supplemental Fig. 19). Although Iba1- and GFAP-positive cells were detected in the area injecting **JigSAP**, the foreign body reaction level was mild and similar to the area injecting **RADA16** which is already used in humans¹⁸. Indeed, NeuN-positive cells were detected near the Iba1-positive area (Supplemental Fig. 20).”

9) Reviewer Comment: “The immunohistochemistry evaluations in this manuscript are completely inadequate for the conclusions made. The authors need to provide more lower and higher magnification images that show the relative differences between groups claimed in the graphs in figure 4. More histological analysis on lesion size differences as well as size and retention of

hydrogels needs to be presented. Furthermore, information about where the IHC quantification is being performed must be provided. i.e. how far away from the non-neural lesion border of the lesion are the neuron counts being performed.”

Response: We appreciate the comments. As reviewer #2 commented, we replaced with new images in Figs. 4 and 5 (we split Fig. 4 into Figs. 4 and 5) as follows.

Figure 4

Figure 5

We also added the lesion size difference measured by the Iba1-positive area (Fig. R2-9; Supplementary Fig. 22). The size was measured by using the volume measurement function of Stereo Investigator (MBF).

Figure R2-9: Evaluation of the lesion size after peptide injection.

a, Iba1 images at the injured area of control. **b**, The volume of the Iba1-positive was measured using the volume measurement function of Stereo Investigator. Scale bar: 500 μ m.

As for hydrogel retention, we could not detect the **JigSAP** hydrogel at 7 days after injection into the injured core. As for the information of IHC quantification at the penumbra, we described in the Cell counting using 2D and 3D stereology section in the material and methods (page 40).

“Volume measurement was performed using the program of Stereo Investigator. For cell counting at the penumbra, the immunopositive cells in the cerebral cortex were counted within 1 mm from the edge of the injured core.”

10) Reviewer Comment: “To claim that the JigSAP material is a superior hydrogel carrier for this stroke treatment a group receiving a benchmarking hydrogel must be included (i.e. comparisons with the RADA hydrogel that are used as comparators in the front end of the paper should be included in the in vivo work).”

Response: We appreciate the comments to improve the quality of the manuscript. In this revised manuscript, we compared **JigSAP** material with **RADA16** materials in the Rose Bengal photothrombotic stroke model (**Fig. R2-10**) and showed that **JigSAP** improved recovery, while **RADA16** did not.

Figure R2-10: Evaluation of the lesion size after peptide injection.

a, Iba1 images at the injured area of control. **b**, The volume of the Iba1-positive was measured using the volume measurement function of Stereo Investigator. Scale bar: 500 μ m.

We added the information to the main text (page 16) as follows and Fig. 6b.

“To compare the effect of **VEGF-JigSAP** with **VEGF-RADA16**, we injected **JigSAP** and **RADA16** peptides incorporating **VEGF-JigSAP** and **VEGF-RADA16**, respectively. In this experiment, to exclude the possibility that **VEGF-JigSAP** +

JigSAP enhanced functional recoveries only in the dMCAO model, we used a photothrombotic model. **VEGF-JigSAP + JigSAP** also improved behavioral recovery in the photothrombotic model, while **VEGF-RADA16 + RADA16** did not (Fig. 6b).”

Minor Issues:

11) Reviewer Comment: “To claim the peptide is cell-adhesive the authors need to show cell morphology rather than just simply DAPI staining for their fibroblast experiments shown in Supplementary Figure 3. This reviewer would like to see a morphology stain, such as Phalloidin actin filament staining, to show the morphology of the cells to support the cell adhesion claims. This experiment needs to be longer term too, the current experiment lasted for only 30 minutes which is inadequate time to assess cell adhesion or cytotoxicity.”

Response: We agree, and to address this issue, we examined cell adhesive, actin remodeling, and cytotoxic properties by comparing them with fibronectin area (Figs. **R2-11, 2-12; Supplementary Figs. 6, 7**). The cell number of **JigSAP** and **RADA16** at 3 and 24 hours after plating was not different from fibronectin. However, their actin remodeling property was significantly different from fibronectin. These results suggest that both **JigSAP** and **RADA16** has cell adhesive property, but they do not activate integrin signal. Although **RADA16** is well-known as cell adhesive supramolecular peptides, RGD peptides that inhibit integrin-binding do not affect cell adhesion of **RADA16** (Zhang et al., Biomaterials 16, 1385-1393, 1995). Thus, both **JigSAP** and **RADA16** has cell adhesive property, but do not activate integrin signal to promote actin remodeling.

Figure R2-11: Cell adhesion property of JigSAP peptide.

a–e, DAPI fluorescent images of fibroblasts attached on a chamber slide coated with Sigmacote (**a**), uncoated control (**b**), Fibronectin (FN) (**c**), **RADA16** (**d**), and **JigSAP** (**e**). Scale bar: 50 μm . **f**, Cell scoring of attached fibroblasts by stereology. $*P < 0.05$. Data are mean \pm SD.

Figure R2-12: Cell morphology cultured on FN, JigSAP, and RADA16.

a–c, Phalloidin (magenta) and DAPI (cyan) fluorescent images of fibroblasts cultured for 24 h on a chamber slide coated with FN (**a**), **JigSAP** (**b**), and **RADA16** (**c**). Scale bar: 50 μm . **d**, Cell scoring of attached fibroblasts by stereology. $*P < 0.05$. Data are mean \pm SD.

We added the information to the main text (page 8).

“**JigSAP** showed cell adhesive properties similar to those for the well-known cell adhesive amphiphilic peptide **RADA16** (Ac-RADARADARADARADA-NH₂), while both **JigSAP** and **RADA16** did not have an actin remodeling property as Fibronectin (Supplementary Figs. 6, 7).”

12) Reviewer Comment: “For all graphs the individual data points must be overlaid on the graphs and individual n values must be included. Information on the statistical tests used must also be included. Currently, none of this is provided. These are requirements outlined by Nature communications for publication.”

Response: We appreciate the comments. We corrected them for publication in *Nature Communications*.

13) Reviewer Comment: “The authors need to clarify how they determined the relative incorporation of tagged versus non-tagged proteins.”

Response: We thank the comment. We described it in the “Sustained-release assay” section in the methods in the initial version. In the revised manuscript, we described the information in the legend of **Fig. 2**, too.

“Figure 2: Efficient incorporation and sustained-release of JigSAP-tagged EGFP protein.

a, Experimental design: the sequence encoding the **JigSAP** peptide was attached to the C-terminal of *EGFP* cDNA. **EGFP-JigSAP**-expressing plasmids were transfected into 293T cells. Partially purified **EGFP-JigSAP** proteins were incubated with **JigSAP** peptides. Ribbon diagram showing the GFP structure was obtained from the literature. **b–d**, The incorporation ratios of peptide-tagged and non-tagged GFPs. **EGFP-JigSAP** (**b**, red), **EGFP-5V** (**c**, blue), and **EGFP-RADA16** (**d**, gray) were incorporated into **JigSAP**, **5V**, and **RADA16** peptide nanofibers, respectively. Non-tagged **EGFP** (**b–d**, black) was used as a control. **Protein incorporation (%) = [(weight of input) – (weight of the soluble fraction)] / (weight of input) × 100 (%)**. **e**, The ratios of released GFP to incorporated GFP. **EGFP- JigSAP** (red), **EGFP-5V** (blue), and **EGFP-RADA16** (gray) were released from **JigSAP**, **5V**, and **RADA16** peptide nanofibers, respectively. **Released protein / incorporated protein (%) = (total weight of the soluble fraction) / (weight of incorporated protein) × 100 (%)**.”

14) Reviewer Comment: “More information on the in vivo experimental methods needs to be included in the methods section such as information on size of needle, rate of injections, anatomical coordinates of injection etc.”

Response: We thank the comment. We added these information and split “**Mouse ischemic stroke model and injection**” into “**Mouse ischemic stroke model**” and “**JigSAP injection into the injured area**” in the method (page 36-38) as follows.

“Mouse ischemic stroke model

Six- to Eight-week-old of C57BL/6J **female** mice were used for ischemic stroke model. Mice were deeply anesthetized by spontaneous inhalation of isoflurane, and temporalis muscle was removed. **dMCAO model was modified from the original method⁵⁸ and described previously⁵⁷. Briefly, the right middle cerebral artery (MCA) was exposed by using an Ideal Micro-Drill (CellPoint Scientific; MD, U.S.A.) and the vessel was cauterized using a small vessel cauterizer (Gemini Cautery Kit, CellPoint Scientific). The hole was located at a position relative to bregma: 1.5 mm posterior and 6.5 mm lateral. The photothrombotic ischemia stroke mouse model was modified from the original method⁵⁹ and prepared by the following methods. After the skin removal, the right side of the skull was covered with the polyvinyl chloride mask with a square whole. The square hole was 3 mm transverse (relative to bregma: 0.5 mm to 3.5 mm) and 5 mm longitudinal (relative to bregma: 3.5 mm to -1.5 mm). At 5 min after intraperitoneal injection of 10 mg/mL of Rose bengal, the exposed square area was illuminated for 10 minutes by a halogen lamp (model L-62; Hozan, Osaka, Japan).**

JigSAP injection into the injured area

At 7 days after dMCAO **or photothrombosis**, the mice were deeply anesthetized by spontaneous inhalation of isoflurane. **Glass capillaries were prepared using a puller (model GD-1; Narishige, Tokyo, Japan) to obtain around 100 μm-tip. The MCA for dMCAO model and the center of the square area for the photothrombotic model was exposed again by tweezers and 2 μL of PBS, 1.0% JigSAP peptides, VEGF-JigSAP (1.6 ng), 1.0% JigSAP peptides mixed with VEGF (1.6 ng), and 1.0% JigSAP peptides mixed with VEGF-JigSAP (1.6 ng) were injected into the ischemic core (1 mm deeper**

from the surface of the brain) for 5 min using microinjector (MO-10; Narishige). For EdU-labeling, 50 $\mu\text{g/g}$ body-weight of EdU was injected intraperitoneally 21 times every 8 h for 7 days.”

15) Reviewer Comment: “More information is needed for the rheology testing including shear rates, strain values, and replication.”

Response: We added detailed information about rheology measurements in the Rheology measurements section in Methods as follows: “Rheology measurements were conducted with rotational rheometer Kinexus lab+ of Malvern Panalytical (Malvern, UK) attached with Peltier plate cartridge and 20 mm ϕ convex plate and 20 mm ϕ parallel plate geometries (PU20 and PLS20). 250 μL of a sample was loaded and sandwiched by the plates with a gap of 0.2 mm. Storage and loss moduli (G' and G'') were obtained by amplitude sweep measurements at 1.0 Hz at strains from 0.1 to 1000% at 20 °C. G' and G'' were taken at plateau values in the linear viscoelastic region. Replication of the rheological data was confirmed by measuring independently-prepared three samples, which showed essentially identical profiles (**Fig. R2-13**). The rheological data for replication study were added to **Supplementary Fig. 28**.

Figure R2-13: Three independent measurements of strain-dependent storage (G' , red) and loss (G'' , blue) modulus profiles of **JigSAP** in DMEM containing 1.0 wt% HEPES at 20 °C for a replication study (incubation time: 48 h, peptide concentration: 1.0 wt%, pH 7.4). G' values at 0.10% shear strain: a) 1.14×10^4 Pa, b) 1.13×10^4 Pa, c) 1.02×10^4 Pa.

16) Reviewer Comment: “Vendor and catalogue numbers are required for all antibodies.”

Response: We thank the comment. We highlighted them in the “Reagents, cells, and mice” section in methods.

Reviewer #3

The authors detail the development of a jigsaw-shaped self-assembling peptide hydrogel that allows for efficient incorporation and sustained release of proteins. The peptides self-assemble into one-dimensional nanostructures in physiological conditions and subsequently form a hydrogel that allows for cell-adhesion to act as an artificial ECM for regenerative therapy. The authors illustrate the efficacy of their JigSP hydrogel in a subacute-chronic phase mouse stroke model, in which the VEGF-loaded hydrogels enhanced injured tissue regeneration. As the authors highlight, supramolecular peptides have been extensively studied for regenerative medicine; however, their system is capable of releasing secreted proteins. The significance of their system and how its design elements impart this key release feature are however not thoroughly explored. Particularly, the advantages of the JigSP design are not clearly highlighted and comparisons made to other self-assembling peptides (RADA16 and 5V) are superficial, dampening the reviewer's enthusiasm.

We would like to thank reviewer #3 for her/his support and helpful comments and suggestions. We have made all of the changes suggested by this reviewer as listed below.

- 1) Reviewer Comment: “It is unclear as to where AXXXA motif is incorporated into the JigSAP design as written in the main text; it needs to be highlighted (Results, ‘Material design and characterization’). Moreover, the identity of the ‘X’-labeled amino acids should be specified (polar, nonpolar, etc.) for this particular motif. The hydrophobic surface stated to be used here (IAMAI) is confusing as it suggests these amino acids are conjugated directly to each other, as opposed to there being hydrophilic amino acids in between these. This should be made more clear in the main text”

Response: In response to the reviewer's suggestions, the AXXXA motif in the **JigSAP** amino acid sequence was italicized for highlighting. The identity of the X-labelled amino acids was also specified as follows (page 5 in the main text): “**JigSAP** has the

sequence Ac-RIDARMRADIR-NH₂ (Fig. 1a, the underlined part indicates the AX¹X²X³A motif; X¹ and X³: polar amino acids, X²: non-polar amino acid), with...”. We also changed the description “IAMA1” to “I–A–M–A–I” to avoid confusion.

2) Reviewer Comment: “The authors need to provide purity data (such as from HPLC) and confirmation of correct molecular weight (such as MALDI-ToF or other mass spectra).”

Response: As purity data, we added analytical HPLC profiles of **JigSAP**, **5V** and **RADA16** used in this study to **Supplementary Fig. 26 (Fig. R3-1)**.

Figure R3-1: High-performance liquid chromatography (HPLC) profiles of a) **JigSAP**, b) **5V** and c) **RADA16** monitored by absorption at 220 nm at 25 °C. The fractions containing each peptide were pointed by arrows. Flow rate: 1.0 mL min⁻¹, Column: YMC-Triart C18 (TA12S05-2546WT), Apparatus: JASCO UV-4075 and PU-4180 HPLC System, Gradient profile: water/acetonitrile = 90/10 (0 to 5 min), 40/60 (30 min), linear gradient between 5 and 30 min.

For confirmation of the molecular weight, MALDI TOF MS data of **JigSAP**, **5V** and **RADA16** used in this study were added to **Supplementary Fig. 27 (Fig. R3-2)**. The isotope patterns of the observed mass signals corresponding to JigSAP, 5V and RADA16 matched their theoretical isotope patterns.

Figure R3-2: Matrix assisted laser desorption/ionization time of flight mass spectrometry (MALDI TOF MS) data of a) **JigSAP**, b) **5V** and c) **RADA16** measured by autoflex speed spectrometer of Bruker with a reflector positive mode using 2,5-dihydroxybenzoic acid (DHB) as a matrix. Theoretical isotope distribution profiles of protonated complexes of d) **JigSAP**, e) **5V** and f) **RADA16** calculated by iMass software (ver. 1.4, Mobile Science Apps).

3) **Reviewer Comment:** “Figure 1C caption says CD spectra represent 20C but the main text states 25C. This needs to be clarified.”

Response: We thank the careful reading. We corrected the temperature written in the main text to 20 °C.

4) **Reviewer Comment:** “Figure 1D is not discussed in the main text after CD results, which is a more logical place to discuss the observation of peaks corresponding to beta sheet character.”

Response: We thank the constructive comment. In response to the reviewer’s suggestion, we revised the discussion about the **JigSAP** hydrogelation process and its

structural characterization. The discussion about the IR spectroscopic results was moved to the place after the CD results.

5) Reviewer Comment: “Figure 1E depicts TEM image of JigSAP nanofibers. The magnification is too low and a higher magnification image should be depicted here. Moreover, the diameter of these structures should be assessed. In addition, no information is provided regarding the sample preparation for the nanofibers in the figure caption. What is the concentration, ageing time, solvent conditions, etc.? Supplementary Figure 1 is a more suitable magnification, but the image is blurry and needs better focusing. Moreover, it appears the filaments have some degree of twisting. The authors should comment on this observed behavior with respect to their JigSAP molecular design, which contrasts with the statement in the main text (Page 7, lines 11-12) about twisting.”

Response: We thank the encouraging comments to improve the quality of the manuscript. In the revised manuscript, we changed the TEM image in **Fig. 1** to a higher magnification image. Images with low magnification are shown in **Supplementary Fig. 1**. The average diameter of the nanofibers (3.2 nm) was described in the revised manuscript. Sample preparation method including sample concentration, aging time, solvent condition and uranyl acetate concentration was described in the figure caption and the Method section as follows.

8244 JigSAP 2-2.tif
17:11:51 4/1/2021
TEM Mode: Imaging

100 nm
HV=100.0kV
Direct Mag: 200000x
#Enter string which describes user's facility or name

Camera: XR16, Exposure(ms): 3200 Gain: 1, Bin: 1
Gamma: 1.00, No Sharpening, Normal Contrast

8244 JigSAP 2.tif
17:08:48 4/1/2021
TEM Mode: Imaging

400 nm
HV=100.0kV
Direct Mag: 50000x
#Enter string which describes user's facility or name

Camera: XR16, Exposure(ms): 3200 Gain: 1, Bin: 1
Gamma: 1.00, No Sharpening, Normal Contrast

“Sample preparation method: TEM observations were conducted with H-7600 of Hitachi (Tokyo, Japan) under 100 kV accelerating voltage. A peptide sample (1.0 wt%) dispersed in aqueous NaHCO₃ (8.8 wt%) solution was diluted with distilled water by 10 times prior to the observation. 5 μL of a sample was placed on a parafilm. Then, a carbon coated 400 mesh copper grid was positioned on top of the drop for 10 s and washed by a droplet of distilled water. For staining, a drop of 2 wt% uranyl acetate was placed on parafilm and the grid was positioned on top of the drop for 10 s. Excess liquid was gently removed using an absorbing paper. After air drying, the grid was submitted to TEM observation.”

“Figure caption: Peptide concentration: 0.10 wt%, aging time: 10 s, solvent condition: 0.88 wt% aqueous solution of NaHCO₃, uranyl acetate concentration: 2.0 wt%.”

As the reviewer points out, the filaments show twisted morphologies, which conflicts with the statement in the original manuscript. Based on this observation, we would like to amend the main text to remove the statement “minimal twisting” from the discussion part about the IR results for more suitable and correct explanation.

6) Reviewer Comment: “Figure 1 should also include CLSM images of 5V hydrogel or the RADA16 hydrogel image should be moved to the supplemental information alongside the other data related to the gel.”

Response: Therefore, we moved the CLSM image of RADA16 hydrogel to **Supplementary Fig. 5**. 5V does not form hydrogel.

7) Reviewer Comment: “TEM images should also be included of the RADA16 peptide assembly. Moreover, the high mag TEM of 5V (Sup. Fig. 8b) is blurry and a more focused image should be included. The sample preparation conditions for these TEM images should be included in the caption (concentration, solvent, age, etc.) and a diameter measurement should be included for the filamentous structures. The concentration of uranyl acetate for TEM imaging should also be included for the methods.”

Response: TEM images of **RADA16** and **5V** with low and high magnification were added to **Supplementary Fig. 12**. Fiber diameters and sample preparation conditions were described in the figure caption as follows.

“Diameters: 3.9 nm (**RADA16**), 3.3 nm (**5V**), peptide concentration: 0.10 wt%, aging time: 10 s, solvent condition: 0.88 wt% aqueous solution of NaHCO₃, uranyl acetate concentration: 2.0 wt%.”

8) **Reviewer Comment:** “For the incorporation of EGFP proteins into the JigSAP, 5V, and RADA16 fibers, were the two mixed with the peptides in a monomeric state or after assembly had occurred. This needs to be clarified, as assembly of the peptides into monomers may impede protein incorporation. Moreover, CD results suggest that it takes 48 hours for the formation of a hydrogel with a strong beta sheet signal at room temperature; however, for the incorporation of protein, the self-assembling peptides and proteins were mixed for 24 h at 37C. There is no experimental evidence provided that suggests that this is adequate conditions for beta sheet and hydrogel formation, which thus influences the amount of protein incorporation and its subsequent release, and thus the data collected here may not be representative of the system.”

Response: To prepare hydrogels incorporating proteins, a peptide and a protein possessing the corresponding peptide tag were mixed as a solution state, in which most

of the peptide molecules were dispersed in monomeric state. The mixture was incubated at 37 °C under 5% CO₂ over **24 h** or longer for gelation. We added the description about this sample preparation procedure to the Method section.

We have performed time-course CD and FI-IR measurements of JigSAP hydrogels. Based on the updated spectroscopic data obtained by the most careful measurements using freshly prepared samples, we would like to amend the CD spectral data shown in Fig. 1g. The time-course rheological profile shows sharp enhancement of G' between 12 and 24 h incubation, and the G' reached plateau after 24 h (Fig. R3-3a). In the time-course CD change, the Cotton effect corresponding to β -sheet structure became dominant after 24 h incubation (Fig. R3-3b). IR spectral change also reached plateau after 24 h incubation (Fig. R3-3c,d). These rheological and spectroscopic analyses indicate that 24 h-incubation after mixing a peptide and a protein in a solution is adequate for sufficient hydrogelation.

Figure R3-3: a) Time-course change in storage modulus G' of JigSAP in DMEM containing 1.0 wt% HEPES at 20 °C at 2, 5, 12, 24, 48, and 100 h after incubation (peptide concentration: 1.0 wt%, pH 7.4). G' values at the shear strain of $10^{-1}\%$ were plotted. b) Circular dichroism (CD) spectra of JigSAP in DMEM containing 1.0 wt% HEPES at 20 °C at 0, 2, 5, 12, 24, 48, and 100 h after incubation (peptide concentration: 1.0 wt%, pH 7.4). c) Time-course changes of the peak position of the amide I band around 1620 cm^{-1} . d) Time-course changes of the fractions of the β -sheet structure (filled circles) and the others (open square) in the amide I band of JigSAP.

9) Reviewer Comment: “Why is there a faster release of EGFP in the JigSP system compared to the RADA16 and 5V system? The differences likely could be related to molecular design elements, supramolecular packing, and critical micellization concentration (CMC) for the studied systems. The authors should discuss this.”

Response: We thank the reviewer for this constructive comment to improve the discussion in our paper. As an important factor that should influence the releasing rate of EGFP, we compared critical aggregation concentrations (CACs) of the peptides. By monitoring fluorescence intensity changes of thioflavin T (ThT) in the presence of a peptide at different concentrations, CACs of **JigSAP** and **5V** were measured to be 0.11 wt% and 0.056 wt%, respectively (**Fig. R3-4**; **Supplementary Fig. 16**). CAC of **RADA16** is reported as 0.023 wt% (Chen, Y. *et al. Int. J. Nanomed.* **13**, 2477–2489 (2018)). Since an amphiphile possessing higher hydrophilicity generally shows a higher CAC value, this result indicates that JigSAP is more hydrophilic than **RADA16** or **5V**. Thus, it is likely that the relatively high hydrophilicity of **JigSAP** allows for the faster protein release than **RADA16** or **5V** systems. We added this discussion to the revised manuscript (page 11).

Figure R3-4: Characterization of critical aggregation concentrations (CACs) of a) **JigSAP** and b) **5V** in DMEM containing 1.0 wt% HEPES at 25 °C by monitoring ThT fluorescence intensity changes. ThT concentration: 25 μ M. Excitation: 440 nm. Fluorescence: 480 nm. The fluorescence spectra were measured with RF-6000 spectrometer of Shimadzu (Kyoto, Japan).

“By monitoring fluorescence intensity changes of thioflavin T (ThT) in the presence of a peptide at different concentrations, CACs of **JigSAP** and **5V** were measured to be

0.11 wt% and 0.056 wt%, respectively (Supplementary Fig. 16, CAC of **RADA16**: 0.023 wt%³⁰). This result indicates that **JigSAP** is more hydrophilic than **5V** or **RADA16**. Thus, it is likely that the relatively high hydrophilicity of **JigSAP** allows for the faster protein release than **RADA16** or **5V** systems.”

10) Reviewer Comment: “In the in vitro studies do not include comparisons of the JigSP system to the RADA16 and 5V systems for VEGF release, thus the advantages of this molecular design are not exemplified in a biomedical application context. These comparisons should be made and discussed to highlight the significance of the JigSP hydrogel system.”

Response: We appreciate the comments to improve the quality of the manuscript. In this revised manuscript, we compared **JigSAP** material with **RADA16** materials in the Rose Bengal photothrombotic stroke model (**Fig. R3-5**) and showed that **JigSAP** improved recovery, while **RADA16** did not.

Figure R3-5: Evaluation of the lesion size after peptide injection.

a, Iba1 images at the injured area of control. **b**, The volume of the Iba1-positive was measured using the volume measurement function of Stereo Investigator. Scale bar: 500 μ m.

We added the information to the main text (page 16) as follows and Fig. 6b.

“To compare the effect of **VEGF-JigSAP** with **VEGF-RADA16**, we injected **JigSAP** and **RADA16** peptides incorporating **VEGF-JigSAP** and **VEGF-RADA16**, respectively. In this experiment, to exclude the possibility that **VEGF-JigSAP** +

JigSAP enhanced functional recoveries only in the dMCAO model, we used a photothrombotic model. **VEGF-JigSAP + JigSAP** also improved behavioral recovery in the photothrombotic model, while **VEGF-RADA16 + RADA16** did not (Fig. 6b).”

11) Reviewer Comment: “The in vivo release rate of tagged proteins from the JigSP hydrogel should be included in this study. The in vitro release rates studied are likely very different than that in vivo, and the release rate will significantly impact the efficacy of the hydrogel system.”

Response: We thank the comment. Since the quantitative analysis for in vivo was difficult, we performed qualitative analysis (**Fig. R3-6**). We injected **JigSAP** and **RADA16** peptides incorporating **EGFP-JigSAP**, **EGFP-RADA16**, respectively, into the non-injured brain and fixed them at 3 and 72 hours after injection. As we expected, **EGFP-RADA16** was clearly detected both at 3 and 72 hours after injection, **EGFP-JigSAP** was clearly detected only in 3 hours after injection. These results suggest that **EGFP-JigSAP** was released and diffused in the brain.

Figure R3-6: EGFP-JigSAP and EGFP-RADA16 injections into the brains.

a-d, EGFP fluorescent images (green) and DAPI (cyan) images of the brains at 3h (b) and 24 h (c, d) after injection. The arrows in (a, b, d) show EGFP signal. Scale bar: 50 μ m.

We added the information to the main text (page 13) as follows and Supplementary Fig. 18.

“To determine whether **JigSAP**-tagged proteins were released from the injected site *in vivo*, we injected **JigSAP** and **RADA16** peptides incorporating **EGFP-JigSAP** and **EGFP-RADA16**, respectively into the non-injured mouse brain (Supplemental Fig. 18). At 3 h after injection, EGFP fluorescent signals were clearly detected at the injected area in both **EGFP-JigSAP** and **EGFP-RADA16** injected brain. In contrast, at 72 h after injection, a clear EGFP signal was only detected in **the EGFP-RADA16** injected brain, suggesting that **EGFP-JigSAP** was released from the injected site.”

Reviewers' Comments:

Reviewer #1:

None

Reviewer #3:

Remarks to the Author:

Review of revised manuscript

Title: Efficient protein incorporation and release by a jigsaw-shaped self-assembling peptide hydrogel for injured brain regeneration

This study by Yaguchi and colleagues describes the development of a new self-assembled peptide hydrogel (called JigSAP) which they use to deliver VEGF to a model of stroke. The authors have revised their manuscript and have included a number of additional experimental studies and associated data sets to address some of my concerns. However, although their efforts at improving the manuscript are admirable, I unfortunately still have considerable issues that remain with this manuscript that are either not addressed and are remaining from the original review or have come out from the newly presented data. I outline these points below using the same headings as the original review of this manuscript. I think these additional points should be addressed before this manuscript could be considered for publication.

Major Issues:

1. Hydrogel gelation – gelation mechanism, time and formulation considerations and their effects on the various outcome measures

i) Gelation mechanism – The authors have performed numerous experiments to address many of my concerns regarding the gelation mechanism. The new CD spectra is more compelling than the previous iteration and demonstrates a likely helix to sheet transition and is accompanied with a good explanation in the text. However, the new FT-IR data seems to somewhat contradict the CD data in that considerable Beta-sheet signal seems to be present in the JigSAP starting at $t=0$ (>60%) and only increases slightly over the time course of gelation. This suggests that assembly of Beta sheets occurs very early and the strengthening of the hydrogel is due to consolidation of Beta-sheet structures and not a dynamic transition from a helix. A sentence or two addressing these data discrepancies would be warranted in the manuscript text.

ii) Rheology – In an attempt to address my concerns, the authors show additional data for G' (storage modulus) changes over the time course of gelation. However, the authors chose not to also show the G'' values, which would have been also measured concurrently during these tests which has perplexed this reviewer. The authors make a point in the manuscript that the “solution turned into a hydrogel after incubation longer than 24 h”. For that statement to be accurate there should be an obvious temporal transition of G'' and G' over the course of gelation and would be associated with a considerable change in phase angle. The data as shown suggests that the hydrogel gets stronger over time but does not show a transition from a sol liquid to a hydrogel over that time course. Rather than showing storage modulus (G') in isolation, the authors should show the $\tan(\delta)$ which is the ratio of G'' to G' for the time points they collected. This measure is a better indication of the transition from a sol to a gel and these data would be need to be included in the manuscript to demonstrate the transition from a solution into a hydrogel as the authors currently claim. Alternatively, the authors could show the cross-over of the G' and G'' traces temporally to show sol-to-gel transition. The rheology with the incorporated EGFP-JigSAP is appropriate (I note that the authors do show G' and G'' here).

iii) Gel usage in in vitro and in vivo experiments – The response here is confusing. If the JigSAP solution is injected immediately, is it already a hydrogel when it is injected? How do you know the in vivo environment is not affecting gelation and its properties?

2. Binding, concentrations and bioactivity of VEGF in hydrogels

iv) The authors have made attempts to address my questions about the faster release of the

modified JigSAP-VEGF compared to the unmodified VEGF but their explanation is not compelling. Presumably the non-specific absorption that they suggest occurs for the non-tagged protein is also occurring for the tagged protein? The authors have not included the hydrogel degradation comparisons as requested.

v) The authors have not shown how the modification of the VEGF with the JigSAP peptide and its incorporation and incubation into hydrogels effects bioactivity as was originally requested. This should be completed.

vi) I still have concerns about the very low concentrations of VEGF used in the in vivo study and I do not find the authors explanation compelling. Minimum concentrations that are effective in vitro are not likely to be overly effective in vivo. A higher concentration of VEGF comparable to what has been effective in other studies should be included. I will add in response to their updated manuscript text on this topic, that making the causal link that the low concentration of VEGF lead to the functional recovery is inappropriate given that there is little to no anatomical evidence provided by the IHC that suggests that such recovery is as a result of the VEGF delivery directly.

3. In vivo elevations

vii) To address my concerns about a lack of foreign body response characterization for this new biomaterial the authors performed a study where they injected JigSAP and a benchmarking control hydrogel into mouse cortex and evaluated the tissue at 72 hours. While admirable that they performed this study, the 72 hour time point is likely too short to adequately evaluate the extent of the foreign body response to the biomaterials (indeed inflammation doesn't peak until around 1 week post injection). Furthermore, the immunohistochemical (IHC) staining observed in these evaluations is extremely poor making interpretations between samples very challenging. The fact that they see no GFAP staining at all in the tissue at 72 hours (inserting a pipet along will upregulate GFAP expression in the tissue) suggests to me that their staining techniques and methods are wholly inadequate and brings in to question all the IHC quantification done on their antibody staining throughout the entire manuscript. Additionally, the NeuN staining provides no reference for where the biomaterial is located.

viii) The immunohistochemistry evaluations in this manuscript remains completely inadequate for the conclusions drawn. The authors need to provide more lower and higher magnification images that show the relative differences between groups claimed in the graphs in figure 4 and 5. Furthermore, information about where the IHC quantification is being performed is still yet to be provided. i.e. how far away from the non-neural lesion border of the lesion are the neuron counts being performed.

ix) The authors have included a benchmarking hydrogel for functional recovery evaluations as requested in the original review. This significant functional recovery after just 7 days after biomaterial injection, given that the lesion volumes in treated groups are not different from controls seems peculiar and difficult to comprehend. Longer evaluations would be required to make the conclusions drawn by the authors.

Minor Issues:

All the minor issues I raised in the original review have been addressed.

Reviewer #4:

Remarks to the Author:

The authors have thoroughly provided responses to the comments raised by the reviewers. The authors have addressed concerns relating to the hydrogel characterization, though more discussion into why and how the JigSAP design elicited better therapeutic effect in the in vivo stroke models should be emphasized, which would increase this reviewer's enthusiasm. The following points below are still of concern with respect to self-assembly and gel characterization, alongside the lack of in-depth discussion mentioned before:

The release rate of VEGF from JigSAP gels was faster for tagged VEGF compared to unmodified and the authors claim that non-specific adsorption onto the gel slows release. Investigation into this phenomenon should be included. For example, Zeta potential measurements of the hydrogel to measure surface charge of the fibers and comparing this to the charge of VEGF (pI) would support electrostatic complexation of VEGF to nanofiber surface. The mitigation of VEGF adsorption through tagging would serve as a clear advantage of the hydrogel system for preserving protein structure and function in addition to controlling the release.

With respect to release rates of protein from JigSAP gels vs RADA16 and 5V, is the observed faster release desirable? JigSAP gels were found to incorporate higher % of added protein, but what was the desired effect with respect to release and/or rheological properties of this design compared to the controls?

The authors included new data and discussion into the kinetics of hydrogel formation, describing the gelation process as 2 stages. In particular, the formation of the gels during the second phase is ascribed to formation of longer fibers during this time. The provided TEM images for this system is of the fibers after 10 seconds of aging based on the figure captions, and not during this later stage. This claim would be easily corroborated by additional TEM analysis of the filaments after longer aging, particularly during this 2nd stage. As presented, the images of fibers included in the manuscript are likely not sufficiently long enough for gelation through physical crosslinking.

Point-by-Point Response to Reviewers

Reviewer #2

This study by Yaguchi and colleagues describes the development of a new self-assembled peptide hydrogel (called JigSAP) which they use to deliver VEGF to a model of stroke. The authors have revised their manuscript and have included a number of additional experimental studies and associated data sets to address some of my concerns. However, although their efforts at improving the manuscript are admirable, I unfortunately still have considerable issues that remain with this manuscript that are either not addressed and are remaining from the original review or have come out from the newly presented data. I outline these points below using the same headings as the original review of this manuscript. I think these additional points should be addressed before this manuscript could be considered for publication.

We thank the Reviewer for her/his constructive comments to improve our manuscript.

Reviewer Comment:

1. Hydrogel gelation – gelation mechanism, time and formulation considerations and their effects on the various outcome measures

i) Gelation mechanism – The authors have performed numerous experiments to address many of my concerns regarding the gelation mechanism. The new CD spectra is more compelling than the previous iteration and demonstrates a likely helix to sheet transition and is accompanied with a good explanation in the text. However, the new FT-IR data seems to somewhat contradict the CD data in that considerable Beta-sheet signal seems to be present in the JigSAP starting at t=0 (>60%) and only increases slightly over the time course of gelation. This suggests that assembly of Beta sheets occurs very early and the strengthening of the hydrogel is due to consolidation of Beta-sheet structures and not a dynamic

transition from a helix. A sentence or two addressing these data discrepancies would be warranted in the manuscript text.

Response: This is indeed an important comment, and we agree with the Reviewer. As the Reviewer points out, the quantitative FT-IR analysis of **JigSAP** shows formation of β -sheet structures with a large fraction (>60%) in the initial stage of the incubation. This result is consistent with the rheological study indicating gel formation of **JigSAP** in the early phase, as we explain in the next comment [ii) Rheology]. Therefore, consolidation of the β -sheet structures should contribute to the strengthening of the hydrogel more largely than the helix-to-sheet transition. We added a description of this point to page 7 in the manuscript text as below.

“Importantly, IR spectroscopic study indicates that **JigSAP** formed the β -sheet structure with larger than 60% fraction even at an initial stage (0 h) and the β -sheet fraction only increased slightly over the incubation time course (Supplementary Fig. 4f). This result suggests that assembly of the β -strands occurs very early and the strengthening of the hydrogel is due to consolidation of the β -sheet structures rather than a dynamic transition from the helix.”

Reviewer Comment:

ii) Rheology – In an attempt to address my concerns, the authors show additional data for G' (storage modulus) changes over the time course of gelation. However, the authors chose not to also show the G'' values, which would have been also measured concurrently during these tests which has perplexed this reviewer. The authors make a point in the manuscript that the “solution turned into a hydrogel after incubation longer than 24 h”. For that statement to be accurate there should be an obvious temporal transition of G'' and G' over the course of gelation and would be associated with a considerable change in phase angle. The data as shown suggests that the hydrogel gets stronger over time but does not show a transition from a sol liquid to a hydrogel over that time course. Rather than showing storage modulus (G') in isolation, the authors should show the $\tan(\delta)$ which is the ratio of G'' to G' for the time points they collected. This measure is a better indication of

the transition from a sol to a gel and these data would be need to be included in the manuscript to demonstrate the transition from a solution into a hydrogel as the authors currently claim. Alternatively, the authors could show the cross-over of the G' and G'' traces temporally to show sol-to-gel transition. The rheology with the incorporated EGFP-JigSAP is appropriate (I note that the authors do show G' and G'' here).

Response: First of all, we would like to apologize for our insufficient presentation of the rheological data in the previous manuscript. In **Figure R2-1**, we show time-course changes in storage modulus G' and loss modulus G'' of **JigSAP**. This result indicates that the G' values are constantly larger than the G'' values from the early phase of the incubation, representing that **JigSAP** forms a hydrogel very early and stiffness of the hydrogel increases over the incubation time course. Therefore, our description “solution turned into a hydrogel” was inaccurate and misleading. In the updated revised manuscript, we deleted this description and revised it as follows (page 5-6):

“Rheological measurements of the hydrated **JigSAP** after 2 h incubation showed a larger storage modulus G' than the corresponding loss modulus G'' , indicating that **JigSAP** readily formed a hydrogel (Fig. 1c, Supplementary Fig. 1). Interestingly, the hydrogel of **JigSAP** showed a sharp enhancement of G' over time from 1.1×10^3 Pa to 1.1×10^4 Pa between 10 and 20 h incubation, visualizing an increase in the hydrogel stiffness (Fig. 1c–e)”.

We also added a time-course change in loss factor ($\tan\delta$) to the **Supplementary Fig. 1 (Fig. R2-1)**.

Figure R2-1: a, b, Time-course changes in (a) strain-dependent storage (G' , red) and loss (G'' , blue) moduli and (b) loss factor ($\tan\delta = G''/G'$) of **JigSAP** in DMEM containing 1.0 wt% HEPES at 20 °C (peptide concentration: 1.0 wt%, pH 7.4).

A rheological measurement of hydrated **JigSAP** incorporating **EGFP-JigSAP** also showed $G' > G''$ relationship over the incubation process, representing hydrogel formation in the early state. Analogous to the **JigSAP** hydrogel, a sharp enhancement of G' was observed during the incubation. These rheological data were added to Supplementary Fig. 16 (Fig. R2-2).

Figure R2-2: a, b, Time-course changes in (a) strain-dependent storage (G' , red) and loss (G'' , blue) moduli and (b) loss factor ($\tan\delta = G''/G'$) of **JigSAP** incorporating **EGFP-JigSAP** in DMEM containing 1.0 wt% HEPES at 20 °C (peptide concentration: 1.0 wt%, **EGFP-JigSAP** concentration: 1.0×10^{-4} wt%, pH 7.4).

Reviewer Comment:

iii) Gel usage in *in vitro* and *in vivo* experiments – The response here is confusing. If the JigSAP solution is injected immediately, is it already a hydrogel when it is injected? How do you know the *in vivo* environment is not affecting gelation and its properties?

Response: We would like to apologize for our insufficient presentation. We injected the soft hydrogel able to inject into the brain as described in [ii)]. To answer the question, we performed the following 2 experiments. One experiment was to dissect and remove the small piece of the injected hydrogel at 48 h after injection (**Fig. R2-3**). The other experiment was the rheological measurements of the **JigSAP** hydrogel formed in the presence of serum (**Fig. R2-4**). G' shows around 10^4 Pa, suggesting that serum does not inhibit **JigSAP** hydrogel formation. Since the *in vivo* hydrogel was too small to perform rheological measurements, we used **Fig. R2-4** in the revised manuscript (**Supplementary figure 9**).

We added the following sentence in the main text (**page 8**).

“JigSAP formed hydrogel in the presence of serum, implicating the application for *in vivo* injection (Supplementary Fig. 9).”

Figure R2-3: 1 μ L of **JigSAP** was injected into the wild-type brain. At 48 h after injection, **JigSAP** hydrogel was dissected and removed. The left panel is the photograph of the whole brain. The white dot square is the area where **JigSAP** was injected. The right panel is the photograph of dissected **JigSAP** hydrogel.

Figure R2-4: a, Photograph of **JigSAP** hydrogel prepared in DMEM containing 1.0 wt% HEPES and incubated with serum for 48 h. b, Strain-dependent storage (G' , red) and loss (G'' , blue) moduli profile of **JigSAP** hydrogel. Peptide concentration: 1.0 wt%, buffer volume: 300 μ L, serum volume: 2.7 mL. The hydrogel was prepared by 48 h incubation at 37 $^{\circ}$ C under 5% CO₂ atmosphere.

Reviewer Comment:

2. Binding, concentrations and bioactivity of VEGF in hydrogels

iv) The authors have made attempts to address my questions about the faster release of the modified **JigSAP-VEGF** compared to the unmodified **VEGF** but their explanation is not compelling. Presumably the non-specific adsorption that they suggest occurs for the non-tagged protein is also occurring for the tagged protein? The authors have not included the hydrogel degradation comparisons as requested.

Response: In the sustained-release measurement, even though **VEGF-JigSAP** is released at significantly higher efficiency than that of unmodified **VEGF**, the release ratio of **VEGF-JigSAP** plateaued halfway (Fig. 3b). Based on the incomplete release of **VEGF-JigSAP** from **JigSAP** hydrogel, it is likely that non-specific adsorption also occurs for the tagged protein. We have added a description of the non-specific adsorption of **VEGF-JigSAP** to page 18 in the manuscript text as follows.

“It should be noted that, even though the proteins tagged with **JigSAP** showed more efficient releasing from **JigSAP** hydrogels than unmodified proteins, their releases plateaued halfway. The incomplete releases suggest that complexation between the tagged-proteins and the peptide assemblies partially contains non-specific adsorption, although the contribution should be smaller than that formed between the non-tagged proteins and the peptides.”

Non-specific adsorption of the peptides onto the gel could be formed by hydrophobic and/or electrostatic interactions. We have measured the Zeta potential of the self-assembled **JigSAP**, indicating +23.3 mV (**Fig. R2-5**). Based on the reported isoelectric point (pI) of VEGF (pI = 8.5)^{Ref1}, VEGF is cationic and thus should be electrostatically repulsive to the **JigSAP** fibers. In contrast, EGFP is negative because of its pI = 5.58^{Ref2}, suggesting that EGFP could form electrostatic complexation with the **JigSAP** fibers. In contrast to the electrochemical properties, a larger amount of unmodified VEGF was incorporated into the **JigSAP** hydrogel than unmodified EGFP, as shown in **Figs. 2b and 3a** (VEGF: 28%, EGFP: <10%). This result suggests that influences of the electrochemical properties of proteins should not be significant in the incorporation process. Importantly, unmodified incorporated VEGF was released much less efficiently than incorporated **VEGF-JigSAP** (VEGF: <10%, VEGF-JigSAP: 50%). Thus, it is likely that the slow and insufficient release of unmodified VEGF should be due to its non-specific adsorption onto the self-assembled **JigSAP** mainly by hydrophobic interaction. It is likely that the tagging with **JigSAP** peptide mitigates adsorption of VEGF with preservation of the protein structure and the capability of releasing. We have added the following discussion to the main text (**page 18-19**).

Ref 1: Houck, K. A., Leung, D. W., Rowland, A. M., Winer, J. & Ferrara, N. Dual regulation of vascular endothelial growth factor bioavailability by genetic and proteolytic mechanisms. *J. Biol. Chem.* **267**, 26031–26037 (1992).

Ref 2: Gasteiger, E. *et al.* Protein Identification and Analysis Tools on the ExPASy Server. in *The Proteomics Protocols Handbook* (ed. Walker, J.) 571–607 (Humana Press, 2005).

“Here, the self-assembled **JigSAP** showed Zeta potential of +23.3 mV (Supplementary Fig. 29). Since VEGF is also cationic ($pI = 8.5$)³⁷, it is likely that the non-specific adsorption is mainly due to hydrophobic interaction rather than electrostatic complexation. The tagging with **JigSAP** peptides should mitigate non-specific adsorption of VEGF onto the peptide assemblies, which serves as an advantage of this hydrogel system for preserving protein structure and controlling the release. It should be also mentioned that the hydrogel degradation rate was hardly influenced by protein incorporation.”

Figure R2-5: Zeta potential profile of **JigSAP** dispersed at 0.10 wt% in DMEM containing 1.0 wt% HEPES at 20 °C. **JigSAP** in a hydrogel state at 1.0 wt% was dispersed into the buffer for 10-fold dilution immediately before the measurement (peptide concentration: 1.0 wt%, pH 7.4).

For comparison of hydrogel degradation, hydrogels (30 μ L) were suspended in DMEM containing 10% serum (470 μ L), and the mixtures were incubated at 37 °C for seven days under shaking at 5 Hz. Three types of hydrogels were investigated: hydrogels made of (1) **JigSAP** alone, (2) a mixture of **JigSAP** and **VEGF**, and (3) a mixture of **JigSAP** and **VEGF-JigSAP** (peptide concentration: 1.0 wt% = 7.1 mM). After incubation, the supernatants were analyzed by HPLC to quantify the concentrations of **JigSAP** molecules dissociated from the hydrogels. For each condition, three samples were prepared and analyzed independently. For all cases, concentrations of **JigSAP** in the supernatants were below 1 μ M. This means that the percentages of **JigSAP** molecules in the supernatants relative to those in the hydrogels

were nearly 0.01 mol%. These results indicate that degradation of hydrogels hardly occurs irrespective of incorporation of proteins or modification of proteins, and thus, the effects of hydrogel degradation should not be the primary factor determining the releasing rates of the proteins.

We added a description about this point to page 11 in the main text as follows,

“Incorporation of proteins hardly influenced the hydrogel degradation rate (Supplementary Fig. 20).”

Figure R2-6: a–j, HPLC traces of (a) **JigSAP** dissolved in buffer (0.010 wt% = 71 μ M, not hydrogelated), (b–d) supernatants of **JigSAP** hydrogels, (e–g) supernatants of hydrogels made of a mixture of **JigSAP** and **VEGF** and (h–j) supernatants of hydrogels made of a mixture of **JigSAP** and **VEGF-JigSAP**. Injection volume: 20 μ L, Detection by absorption at 220 nm at 25 $^{\circ}$ C, Flow rate: 1.0 mL min^{-1} , Column: YMC-Triart C18 (TA12S05-2546WT), Apparatus: JASCO UV-4075 and PU-4180 HPLC System, Gradient profile: water/acetonitrile = 90/10 (0 to 5 min), 40/60 (30 min), linear gradient between 5 and 30 min.

Reviewer Comment:

v) The authors have not shown how the modification of the VEGF with the JigSAP peptide and its incorporation and incubation into hydrogels effects bioactivity as was originally requested. This should be completed.

Response: We would like to apologize for not presenting the data. We confirmed that the modification of **JigSAP** and the incorporation into and the releasing from the **JigSAP** hydrogel does not affect VEGF bioactivity, as shown in **Fig. R2-7**. HUVECs were cultured with **VEGF-JigSAP** before incorporation, the released **VEGF-JigSAP** from the hydrogel, or a commercially available recombinant VEGF. Endothelial-cell proliferation was evaluated by BrdU ELISA. We added the following description to **page 13** in the main text as follows,

“It should be noted that neither **JigSAP** modification to VEGF nor VEGF incorporation into and releasing from **JigSAP** hydrogel affected VEGF bioactivity (Supplementary Fig. 21).”

Figure R2-7: HUVEC proliferation assay with **VEGF-JigSAP** before incorporation into **JigSAP** hydrogel (**VEGF-JigSAP** before), **VEGF-JigSAP** after release from **JigSAP** hydrogel (**VEGF-JigSAP** after), or commercially available recombinant VEGF. The “**VEGF-JigSAP** after” was obtained from the supernatant of sustained-release assay after 7 days. The concentration of “**VEGF-JigSAP** before” and “**VEGF-JigSAP** after” was determined by VEGF ELISA. HUVECs were plated on a collagen-coated 96-well plate (2.5×10^3 cells/well) in 100 μ l Medium-199 with 0.1% BSA in the presence of “**VEGF-JigSAP** before,” “**VEGF-JigSAP** after,” or recombinant mouse VEGF 164 (493-MV; R&D). Two days after plating, BrdU was added, and the cells were cultured for an additional 24 h. BrdU incorporation was quantified using the BrdU Cell Proliferation ELISA Kit (ab126556; Abcam).

Reviewer Comment:

vi) I still have concerns about the very low concentrations of VEGF used in the in vivo study and I do not find the authors explanation compelling. Minimum concentrations that are effective in vitro are not likely to be overly effective in vivo. A higher concentration of VEGF comparable to what has been effective in other studies should be included. I will add in response to their updated manuscript text on this topic, that making the causal link that the low concentration of VEGF lead to the functional recovery is inappropriate given that there is little to no anatomical evidence provided by the IHC that suggests that such recovery is as a result of the VEGF delivery directly.

Response: First of all, we would like to apologize for our insufficient presentation of IHC. As described below in the comment on [viii)], the new IHC images would eliminate the Reviewer’s concern. Second, we could not prepare μ g order of **VEGF-**

JigSAP. In the current study, we obtained **VEGF-JigSAP** from the supernatant of transiently transfected 293T cells and do not have a stable transfectant which is essential to obtain a high amount of **VEGF-JigSAP**. Since we could not compare the higher concentration of **VEGF-JigSAP** used in the previous study, we deleted the following sentence in the revised manuscript.

The following description was deleted in the revised manuscript “Recently, Nih et al. reported that the single injection of heparin nanoparticles binding 200 ng of VEGF promotes functional recoveries. However, in the current study, we injected **JigSAP** peptides incorporating 1.6 ng of **VEGF-JigSAP** and observed functional recoveries, suggesting that a lower dose of VEGF may be enough for functional recoveries if VEGF is slowly released from the injected site.”

Reviewer Comment:

3. In vivo elevations

vii) To address my concerns about a lack of foreign body response characterization for this new biomaterial the authors performed a study where they injected JigSAP and a benchmarking control hydrogel into mouse cortex and evaluated the tissue at 72 hours. While admirable that they performed this study, the 72 hour time point is likely too short to adequately evaluate the extent of the foreign body response to the biomaterials (indeed inflammation doesn't peak until around 1 week post injection). Furthermore, the immunohistochemical (IHC) staining observed in these evaluations is extremely poor making interpretations between samples very challenging. The fact that they see no GFAP staining at all in the tissue at 72 hours (inserting a pipet along will upregulate GFAP expression in the tissue) suggests to me that their staining techniques and methods are wholly inadequate and brings in to question all the IHC quantification done on their antibody staining throughout the entire manuscript. Additionally, the NeuN staining provides no reference for where the biomaterial is located.

Response: We would like to apologize for our insufficient presentation of IHC. In the revised manuscript, we fixed the injected mice at 7 days after injection. The new figures are now in supplementary figures 24 and 25.

Figure R2-8: Evaluation of the foreign body responses of **JigSAP** and **RADA16** after brain injection. **JigSAP** and **RADA16** bearing a 4-pentynoyl group (**Alkyne-JigSAP** and **Alkyne-RADA16**) were used to visualize SAP *in vivo*. 1% **JigSAP** conjugated with 0.01% **Alkyne-JigSAP**, and 1% **RADA16** conjugated with 0.01% **Alkyne-RADA16**, or PBS was injected into the non-injured brain. The injected peptide was visualized by the Huisgen 1,3-dipolar cycloaddition reaction with Alexa 647-azide. At 7 days after injection, these mice were fixed. Activated astrocyte marker GFAP (green) (a-d), microglial marker Iba1 (magenta) (e-h), and merged SAP (yellow) images (i-l). Scale bar: 200 μ m.

Figure R2-9: NeuN staining after **JigSAP** or **RADA16** injection. **JigSAP** or **RADA16** were injected into the non-injured brain. At 7 d after injection, these mice were fixed. **a, b**, Neuron marker NeuN (green) images. Scale bar: 200 μ m.

Reviewer Comment:

viii) The immunohistochemistry evaluations in this manuscript remains completely inadequate for the conclusions drawn. The authors need to provide more lower and higher magnification images that show the relative differences between groups claimed in the graphs in figure 4 and 5. Furthermore, information about where the IHC quantification is being performed is still yet to be provided. i.e. how far away from the non-neural lesion border of the lesion are the neuron counts being performed.

Response: We would like to apologize for our insufficient presentation of IHC. We updated the IHC images of Fig. 4 and 5. The detailed descriptions of the IHC quantification are provided in the figure legends.

“For cell counting, Laminin-positive cells were counted from the non-neural lesion borders to 200 μ m away.” “For cell counting, NeuN- and FJC-positive cells were counted from the non-neural lesion borders to 200 μ m away.”

Figure 4

Figure 5

Reviewer Comment:

ix) The authors have included a benchmarking hydrogel for functional recovery evaluations as requested in the original review. This significant functional recovery after just 7 days after biomaterial injection, given that the lesion volumes in treated groups are not different from controls seems peculiar and difficult to comprehend. Longer evaluations would be required to make the conclusions drawn by the authors.

Response: We appreciated the critical comments. We evaluated at 14 and 21 days after injection (Fig. R2-10). We noted that the control mice improved motor function and did

not show the difference with the mice injected VEGF-JigSAP + JigSAP at 21 days after injection. The following description was added to the main text on page 16.

“The behavioral recovery effect by **VEGF-JigSAP + JigSAP** was observed at 14 days after injection, but not at 21 days after injection because of the recovery of control mice (Supplementary Fig. 28).”

Figure R2-10: Ratios of fault steps to total steps during FFT at 14 and 21 days after injection. * $P < 0.05$. $n = 7$. Data are mean \pm SEM.

Reviewer #3

The authors have thoroughly provided responses to the comments raised by the reviewers. The authors have addressed concerns relating to the hydrogel characterization, though more discussion into why and how the JigSAP design elicited better therapeutic effect in the in vivo stroke models should be emphasized, which would increase this reviewer's enthusiasm. The following points below are still of concern with respect to self-assembly and gel characterization, alongside the lack of in-depth discussion mentioned before:

Response: We thank the Reviewer for her/his helpful suggestions to improve our manuscript.

Reviewer Comment:

The release rate of VEGF from JigSAP gels was faster for tagged VEGF compared to unmodified and the authors claim that non-specific adsorption onto the gel slows release. Investigation into this phenomenon should be included. For example, Zeta potential measurements of the hydrogel to measure surface charge of the fibers and comparing this to the charge of VEGF (pI) would support electrostatic complexation of VEGF to nanofiber surface. The mitigation of VEGF adsorption through tagging would serve as a clear advantage of the hydrogel system for preserving protein structure and function in addition to controlling the release.

Response: Non-specific adsorption of the peptides onto the gel could be formed by hydrophobic and/or electrostatic interactions. We have measured the Zeta potential of the self-assembled **JigSAP**, indicating +23.3 mV (**Fig. R2-5**). Based on the reported isoelectric point (pI) of VEGF (pI = 8.5)^{Ref 1}, VEGF is cationic and thus should be electrostatically repulsive to the **JigSAP** fibers. In contrast, EGFP is negative because of its pI = 5.58^{Ref 2}, suggesting that EGFP could form electrostatic complexation with the **JigSAP** fibers. In contrast to the electrochemical properties, a larger amount of unmodified VEGF was incorporated into the **JigSAP** hydrogel than unmodified EGFP, as shown in **Figs. 2b and 3a** (VEGF: 28%, EGFP: <10%). This result suggests that

influences of the electrochemical properties of proteins should not be significant in the incorporation process. Importantly, unmodified incorporated VEGF was released much less efficiently than incorporated **VEGF-JigSAP** (VEGF: <10%, VEGF-JigSAP: 50%). Thus, it is likely that the slow and insufficient release of unmodified VEGF should be due to its non-specific adsorption onto the self-assembled **JigSAP** mainly by hydrophobic interaction. It is likely that the tagging with **JigSAP** peptide mitigates adsorption of VEGF with preservation of the protein structure and the capability of releasing. We have added the following discussion to the main text (page 18-19).

Ref 1: Houck, K. A., Leung, D. W., Rowland, A. M., Winer, J. & Ferrara, N. Dual regulation of vascular endothelial growth factor bioavailability by genetic and proteolytic mechanisms. *J. Biol. Chem.* **267**, 26031–26037 (1992).

Ref 2: Gasteiger, E. *et al.* Protein Identification and Analysis Tools on the ExPASy Server. in *The Proteomics Protocols Handbook* (ed. Walker, J.) 571–607 (Humana Press, 2005).

“Here, the self-assembled **JigSAP** showed Zeta potential of +23.3 mV (Supplementary Fig. 29). Since VEGF is also cationic ($pI = 8.5$)³⁷, it is likely that the non-specific adsorption is mainly due to hydrophobic interaction rather than electrostatic complexation. The tagging with **JigSAP** peptides should mitigate non-specific adsorption of VEGF onto the peptide assemblies, which serves as an advantage of this hydrogel system for preserving protein structure and controlling the release. It should be also mentioned that the hydrogel degradation rate was hardly influenced by protein incorporation.”

Figure R3-1: Zeta potential profile of **JigSAP** dispersed at 0.10 wt% in DMEM containing 1.0 wt% HEPES at 20 °C. **JigSAP** in a hydrogel state at 1.0 wt% was dispersed into the buffer for 10-fold dilution immediately before the measurement (peptide concentration: 1.0 wt%, pH 7.4).

Regarding the bioactivity of **VEGF-JigSAP** compared with non-tagged VEGF, there was no significant difference (**Fig. R3-2** and **Supplementary Fig. 21**).

Figure R3-2: HUVEC proliferation assay with **VEGF-JigSAP** before incorporation into **JigSAP** hydrogel (**VEGF-JigSAP** before), **VEGF-JigSAP** after release from **JigSAP** hydrogel (**VEGF-JigSAP** after), or commercially available recombinant VEGF. The “**VEGF-JigSAP** after” was obtained from the supernatant of sustained-release assay after 7 days. The concentration of “**VEGF-JigSAP** before” and “**VEGF-JigSAP** after” was determined by VEGF ELISA. HUVECs were plated on a collagen-coated 96-well plate (2.5×10^3 cells/well) in 100 μ l Medium-199 with 0.1% BSA in the presence of “**VEGF-JigSAP** before”, “**VEGF-JigSAP** after”, or recombinant mouse VEGF 164 (493-MV; R&D). Two days after plating, BrdU was added, and the cells were cultured for an additional 24 h. BrdU incorporation was quantified using the BrdU Cell Proliferation ELISA Kit (ab126556; Abcam).

Reviewer Comment:

With respect to release rates of protein from JigSAP gels vs RADA16 and 5V, is the observed faster release desirable? JigSAP gels were found to incorporate higher % of added protein, but what was the desired effect with respect to release and/or rheological properties of this design compared to the controls?

Response: We appreciate the critical comments on the concept of our molecular design. If cell adhesion molecules are used as an artificial ECM, self-assembling peptides (SAPs) should not release the cell adhesion molecules. In contrast, soluble growth factors are used as an artificial ECM, SAPs should release the growth factors. However, even in the research field of developmental biology, the diffusion rate of growth factors remains unknown. Moreover, diffusion rates are different in each growth factor and extracellular environmental factor. Thus, it could be difficult to generalize the desirable release rate. In our manuscript, **VEGF-RADA16** and **VEGF** (without JigSAP tag) + **JigSAP** did not promote functional recovery, suggesting that the release of **VEGF-JigSAP** from **JigSAP** hydrogel was sufficient to promote functional recovery. Future studies evaluating different tissues and different injury models could be important to generalize the desirable release rate.

Reviewer Comment:

The authors included new data and discussion into the kinetics of hydrogel formation, describing the gelation process as 2 stages. In particular, the formation of the gels during the second phase is ascribed to formation of longer fibers during this time. The provided TEM images for this system is of the fibers after 10 seconds of aging based on the figure captions, and not during this later stage. This claim would be easily corroborated by additional TEM analysis of the filaments after longer aging, particularly during this 2nd stage. As presented, the images of fibers included in the manuscript are likely not sufficiently long enough for gelation through physical crosslinking.

Response: We have added a TEM image of **JigSAP** after 24 h aging to **Supplementary Fig. 3b (Fig. R3-3)**. The TEM image visualizes well-developed fibers over 5–10 micrometers lengths forming a network structure. A TEM image of **JigSAP** after 10 s aging shows the formation of fibers shorter than 1-micrometer length (**Fig. R3-4, Supplementary Fig. 3c**).

Figure R3-3: TEM image of **JigSAP** peptide after 24 h aging. Peptide concentration: 0.10 wt%, aging time: 24 h, solvent condition: 0.88 wt% aqueous solution of NaHCO_3 , uranyl acetate concentration: 2.0 wt%. Scale bar: 1.0 μm .

Figure R3-4: TEM image of **JigSAP** peptide after 10 s aging. Peptide concentration: 0.10 wt%, aging time: 10 s, solvent condition: 0.88 wt% aqueous solution of NaHCO_3 , uranyl acetate concentration: 2.0 wt%. Scale bar: 1.0 μm .

Reviewers' Comments:

Reviewer #3:

Remarks to the Author:

The authors have diligently addressed the major issues I identified as part of my two previous reviews of this manuscript. The additional data and updated interpretations presented have addressed all my concerns and have greatly improved the quality and impact of the manuscript. They should be commended for their hard work. I have no further issues that need to be addressed with this work at this time.

Reviewer #4:

Remarks to the Author:

The authors have done a good job in further improving their manuscript. There are no more questions.

Point-by-Point Response to Reviewers (final revision)

Reviewer #3

The authors have diligently addressed the major issues I identified as part of my two previous reviews of this manuscript. The additional data and updated interpretations presented have addressed all my concerns and have greatly improved the quality and impact of the manuscript. They should be commended for their hard work. I have no further issues that need to be addressed with this work at this time.

Response: We thank the Reviewer for her/his kind comments.

Reviewer #4

The authors have done a good job in further improving their manuscript. There are no more questions..

Response: We thank the Reviewer for her/his kind comments.